# RLAIF: Scaling Reinforcement Learning from Human Feedback with AI Feedback

## Abstract

Reinforcement learning from human feedback (RLHF) has proven effective in aligning large language models (LLMs) with human preferences. However, gathering high-quality human preference labels can be a time-consuming and expensive endeavor. RL from AI Feedback (RLAIF), introduced by Bai et al., offers a promising alternative that leverages a powerful off-the-shelf LLM to generate preferences in lieu of human annotators. Across the tasks of summarization, helpful dialogue generation, and harmless dialogue generation, RLAIF achieves comparable or superior performance to RLHF, as rated by human evaluators. Furthermore, RLAIF demonstrates the ability to outperform the supervised fine-tuned baseline even when the LLM preference labeler is of the same size as the policy. In another experiment, directly prompting the LLM for reward scores achieves superior performance to the canonical RLAIF setup, where LLM preference labels are distilled into a reward model. Finally, we conduct extensive studies on techniques for generating aligned AI preferences. Our results suggest that RLAIF can achieve human-level performance, offering a potential solution to the scalability limitations of RLHF.

## 1 Introduction

Reinforcement Learning from Human Feedback (RLHF) is an effective technique for aligning language models to human preferences (Stiennon et al., 2020; Ouyang et al., 2022). It is cited as one of the key drivers of success in modern conversational language models such as ChatGPT (Liu et al., 2023) and Bard (Manyika, 2023). Training language models with reinforcement learning (RL) enables optimization on complex, sequence-level objectives that are not easily differentiable and therefore ill-suited for traditional supervised fine-tuning (SFT).

One obstacle for employing RLHF at scale is its dependence on high-quality human preference labels. This raises the question of whether artificially generated labels can be a viable substitute. Generating labels with large language models (LLMs) is one promising approach, as LLMs have shown a high degree of alignment with human judgment (Gilardi et al., 2023; Ding et al., 2023). Bai et al. (2022b) was the first effort to explore Reinforcement Learning from AI Feedback (RLAIF)[1], where RL was conducted using a reward model trained on LLM preferences. They showed that utilizing a hybrid of human and AI preferences, in conjunction with their "Constitutional AI" self-revision technique, outperforms supervised fine-tuning for training a conversational assistant aligned with human preferences. However, it did not directly compare the efficacy of human vs. AI feedback, leaving the question of whether RLAIF can be a suitable alternative to RLHF unanswered.

In this work, we study the impact of RLAIF and RLHF (see Figure 2) on three text generation tasks: summarization, helpful dialogue generation, and harmless dialogue generation. For summarization and helpful dialogue generation, our experiments show that RLAIF and RLHF are preferred by humans over the SFT baseline 71% and 73% of the time for summarization and 63% and 64% of the time for helpful dialogue generation, respectively, where the differences between RLAIF and RLHF win rates are not statistically significant. We also conduct a head-to-head comparison of RLAIF

---

[1]This is distinct from "Constitutional AI", which improves upon a supervised learning model through iteratively asking a LLM to generate better responses according to a constitution. Both were introduced in Bai et al. (2022b) and are sometimes conflated.

Figure 1: Human evaluators strongly prefer RLAIF and RLHF over the SFT baseline for summarization and helpful dialogue generation. The differences in win rates w.r.t. SFT are not statistically significant. Furthermore, when compared head-to-head, RLAIF is equally preferred to RLHF. For harmless dialogue generation, RLAIF outperforms RLHF.

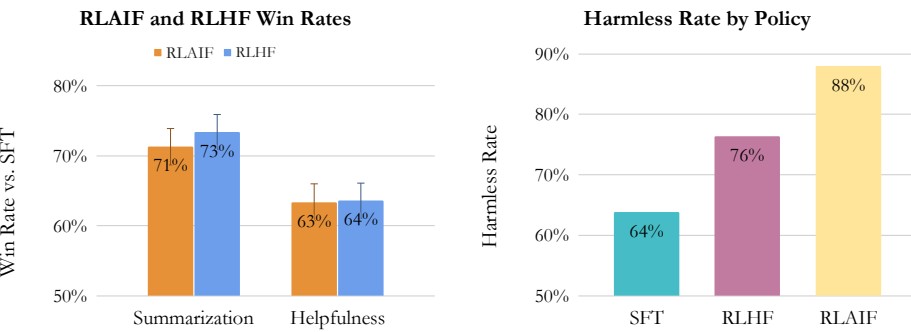

against RLHF and find that both policies are equally preferred[2]. For harmless dialogue generation, human evaluators were tasked with rating the harmlessness of each response independently. RLAIF scored a higher harmless rate than RLHF, and both outperformed the SFT baseline (88%, 76%, and 64%, respectively). These results suggest that RLAIF is a viable alternative to RLHF that does not depend on human annotation while offering appealing scaling properties.

Additionally, we investigate two related questions. First, we explore whether RLAIF can improve upon a SFT policy when the LLM labeler has the same number of parameters as policy. Our results show that even in this scenario, RLAIF improves over the SFT baseline, achieving a win rate of 68%. Second, we conduct an experiment where the off-the-shelf LLM is directly prompted for reward scores during RL, bypassing the step of distilling LLM preference labels into a separate reward model. This method achieves an even higher win rate over SFT than the canonical distillation method.

Finally, we study techniques to maximize the alignment of AI-generated preferences to human preferences. We find that soliciting chain-of-thought reasoning (Wei et al., 2022) consistently improves alignment, while the benefits of using a detailed preamble and few-shot prompting are task-specific. We also conduct scaling experiments to examine the trade-offs between the size of the LLM labeler and alignment with human preferences.

Our main contributions are as follows:

1. We demonstrate that RLAIF achieves comparable or superior performance to RLHF on the tasks of summarization, helpful dialogue generation, and harmless dialogue generation.
2. We show that RLAIF can improve upon a SFT policy even when the LLM labeler is the same size as the policy.
3. We find that directly prompting the LLM for reward scores during RL can outperform the canonical setup where a reward model is trained on LLM preferences.
4. We compare various techniques for generating AI labels and identify optimal settings for RLAIF practitioners.

## 2 METHODOLOGY

In this section, we describe the techniques used to generate preference labels with a LLM, how we conduct RL, and evaluation metrics. Preliminaries on RLHF are provided in Appendix A.

### 2.1 PREFERENCE LABELING WITH LLMS

We annotate preferences among pairs of candidates with an "off-the-shelf" LLM - a model pre-trained or instruction-tuned (Wei et al., 2021) for general usage but not fine-tuned for a specific downstream

---

[2]The win rate for one policy vs. the other is not statistically significantly different from 50%

Figure 2: A diagram depicting RLAIF (top) vs. RLHF (bottom)

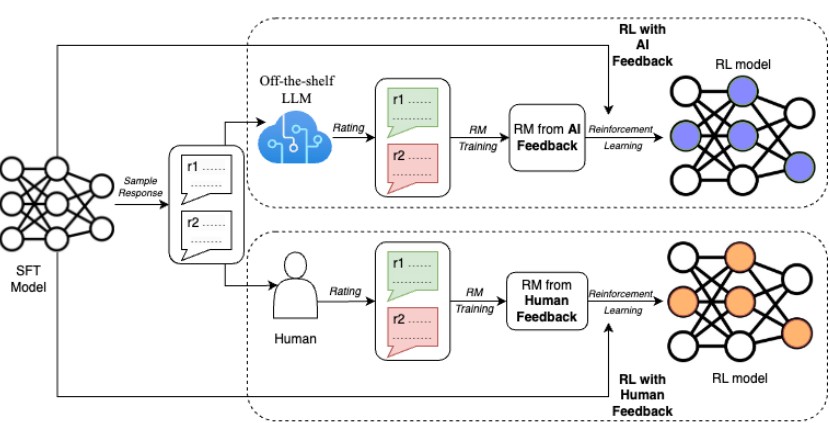

task. Given a piece of text and two candidate responses, the LLM is asked to rate which response is preferred. The prompt is structured as follows (examples in Tables 14 and 20):

1. *Preamble* - Introduction and instructions describing the task at hand
2. *Few-shot exemplars (optional)* - An example input context, a pair of responses, a chain-of-thought rationale (if applicable), and a preference label
3. *Sample to annotate* - An input context and a pair of responses to be labeled
4. *Ending* - Ending text to prompt the LLM (e.g. "*Preferred Response=*")

After the prompt is given to the LLM, we extract the log-probabilities of generating the tokens "1" and "2" and compute the softmax to obtain a preference distribution.

There are numerous alternatives to obtain preference labels from LLMs, such as decoding a free-form response from the model and extracting the preference heuristically (e.g. *"The first response is better"*), or representing the preference distribution as a one-hot representation. However, we choose to use the log-probabilities of generating "1" and "2" because it is straightforward to implement and conveys more information than a one-hot representation through distributed preference distributions.

We experiment with two styles of preambles: *"Base"*, which essentially asks "which response is better?", and *"Detailed"*, which resembles detailed rating instructions that would be given to the human preference annotators (see Table 15 for preambles used for the summarization task). We also experiment with in-context learning, where exemplars were hand-selected to be high-quality and to cover different topics.

### 2.1.1 ADDRESSING POSITION BIAS

The order in which candidates are shown to the LLM can bias which candidate it prefers (Pezeshkpour and Hruschka, 2023; Wang et al., 2023). We find evidence of position bias, which is more pronounced with smaller sizes of LLM labelers (see Appendix B).

To mitigate position bias in preference labeling, we make two inferences for every pair of candidates, where the order in which candidates are presented to the LLM is reversed for the second inference. The results from both inferences are then averaged to obtain the final preference distribution.

### 2.1.2 CHAIN-OF-THOUGHT REASONING

We experiment with eliciting chain-of-thought (CoT) reasoning from our AI labelers to improve alignment with human preferences (Wei et al., 2022). We replace the *Ending* of the standard prompt (e.g. "*Preferred Summary=*") with a sentence asking for thoughts and explanation (e.g. "*Consider the coherence, accuracy, coverage, and overall quality of each summary and explain which one is better. Rationale:*") and then decode a response from the LLM. Finally, we concatenate the original prompt, the response, and the original *Ending* string together, and follow the scoring procedure in Section 2.1 to obtain a preference distribution. See Figure 3 for an illustration.

Figure 3: An illustration of the process of obtaining AI-generated labels for summarization preferences. The LLM is first prompted to explain its thoughts on the quality of the two candidates (blue). The LLM's response is then appended to the original prompt (orange) and fed to the LLM a second time to generate a preference distribution over "1" vs. "2" based on their log-probabilities (green).

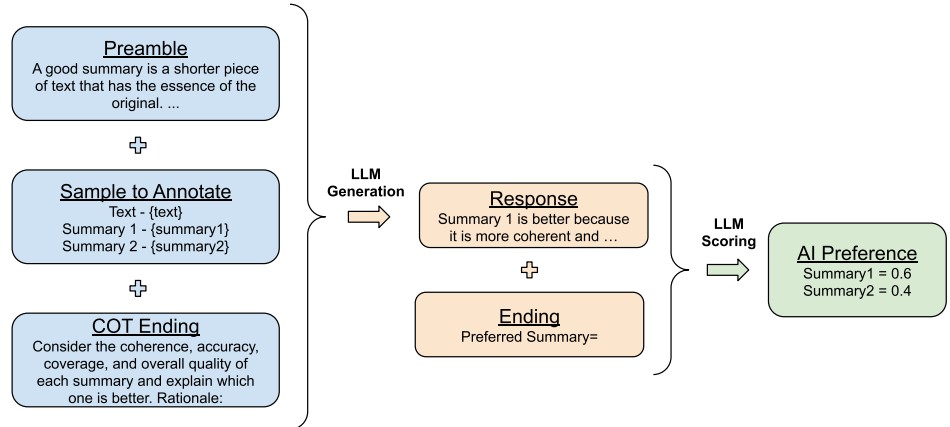

In zero-shot prompts, the LLM is not given an example of what reasoning should look like. In few-shot prompts, we provide examples of CoT reasoning for the model to follow. See Tables 16 and 17 for examples.

## 2.2 REINFORCEMENT LEARNING FROM AI FEEDBACK

After labeling preferences with a LLM, a reward model (RM) is trained to predict preferences. Since our approach produces soft labels (e.g. $[0.6, 0.4]$), we apply a cross-entropy loss to the softmax of the reward scores generated by the RM. The softmax is used to convert the unbounded scores from the RM into a probability distribution.

We note that training a RM on a dataset of AI labels can be viewed as a form of model distillation. We also explore an alternative approach where AI feedback is used directly as the reward signal in RL (see Section 4.2). The latter is much more computationally expensive than the former when the AI labeler is larger than the RM.

Finally, we conduct reinforcement learning to train the RLAIF policy model, using the trained RM to score generations and assign rewards.

## 2.3 EVALUATION

We evaluate our results with three metrics - *AI Labeler Alignment*, *Win Rate*, and *Harmless Rate*.

*AI Labeler Alignment* measures the accuracy of AI-labeled preferences with respect to human preferences. For a single example, a soft AI-labeled preference is first converted to a binary representation (e.g. $[0.6, 0.4] \rightarrow [1, 0]$). Then, it receives a 1 if the label agrees with the human preference and 0 otherwise. The alignment accuracy $z_{acc}$ can be expressed as follows:

$$z_{acc} = \frac{1}{D} \sum_{i=1}^{D} \mathbb{1}[\arg\max_{j} P_{i,j}^{AI} = p_i^H],$$

where $D$ is the preference dataset size, $P^{AI} \in \mathbb{R}^{D \times 2}$ is the matrix of soft AI preferences, and $p^{human} \in \mathbb{R}^D$ is the corresponding vector of human preferences, containing elements 0 or 1 to denote whether the first or second response is preferred, respectively.

*Win Rate* evaluates the end-to-end quality of two policies by measuring how often one policy is preferred by humans over another. Given an input and two generations, human annotators select which generation they prefer according to given guidelines. The percentage of instances where policy

$A$ is preferred over policy $B$ is referred to as the *"Win Rate of A vs. B"*. A 50% Win Rate indicates that $A$ and $B$ are equally preferred.

*Harmless Rate* measures the percentage of responses that are considered harmless or safe by human evaluators. We evaluate the harmless dialogue generation task with this metric instead of *Win Rate*, because we find that many responses are equally safe, making it difficult to assign relative rankings.

## 3 EXPERIMENTAL DETAILS

### 3.1 DATASETS

We use the following datasets for our experiments:

- Reddit TL;DR (Stiennon et al., 2020) - posts from Reddit[3] accompanied by summaries of the posts.
- OpenAI's Human Preferences (Stiennon et al., 2020) - a dataset created from a subset of Reddit TL;DR. Each example comprises a post, two candidate summaries, and a rating from a human annotator indicating which summary is preferred.
- Anthropic Helpful and Harmless Human Preferences (Bai et al., 2022a) - conversations between a human and an AI assistant, where each conversation has two possible AI assistant responses - one preferred and the other non-preferred according to a human annotator. Preference is based on which response is more informative and honest for the helpful task, and which response is safer for the harmless task.

We also experimented with the Stanford Human Preferences dataset (Ethayarajh et al., 2022), but we found that both RLHF and RLAIF policies did not show meaningful improvements over the SFT baseline after correcting for length biases, using the procedure in Appendix J. More dataset details can be found in Appendix C.

### 3.2 LLM LABELING

To enable faster experiment iteration when evaluating AI labeling techniques, we randomly sampled a subset from the training split of each preference dataset, yielding roughly 3-4k examples for each task[4]. For summarization, we further filtered the data to include only examples where human annotators preferred one summary over the other with high confidence[5].

We use PaLM 2 (Google et al., 2023) as our LLM for labeling preferences. The versions we use are instruction-tuned but not previously trained with RL. Unless otherwise specified, we generate AI labels using PaLM 2 Large (L) with the best-performing prompt in Section 4.4. For more details on LLM labeling, see Appendix D.

### 3.3 MODEL TRAINING

All SFT models are initialized from PaLM 2 Extra-Small (XS). For summarization, we fine-tune on the Reddit TL;DR dataset. For all other tasks, we utilize an instruction-tuned variant of PaLM 2 in lieu of task-specific fine-tuning.

RMs are also derived from PaLM 2 XS. RMs are fine-tuned on the full training split of the corresponding preference dataset, where the label is the AI labeled preference for AI feedback RMs and the original human preference label in the dataset for human feedback RMs. We report RM accuracies in Appendix G.

In the RL phase, we train the policy with a modified version of REINFORCE (Williams, 1992) adapted to the language modeling domain. While many recent works use Proximal Policy Optimization

---

[3]`www.reddit.com`

[4]We sample 15%, 10%, and 10% of the training splits for summarization, helpful dialogue generation, and harmless dialogue generation, respectively.

[5]This follows the evaluation procedure in Stiennon et al. (2020). Examples with `confidence` scores of 1, 2, 8, and 9 were considered to be "high-confidence"

(PPO) (Schulman et al., 2017) - a related method that adds a few techniques to make training more conservative and stable (e.g. clipping the objective function), we use REINFORCE with a baseline given that it is simpler yet still effective for the problem at hand. Both policy and value models are initialized from the SFT model. For summarization, we roll out our policy on the training split of the Reddit TL;DR dataset. For the helpful and harmless tasks, we use the training splits of the preference datasets as our initial states. For summarization, we perform simple post-processing on responses generated by post-RL policies as described in Appendix E.

For additional details, see Appendix F for the RL formulation and Appendix G for model training.

## 3.4 HUMAN EVALUATION

For experiments evaluated by win rates, evaluators were presented with an input context and multiple responses generated from different policies (e.g. RLAIF, RLHF, and SFT). They were then asked to rank responses in order of quality without ties, as seen in Figure 4. Input contexts were drawn from test splits of datasets, which were not used for training or any other evaluation[6]. Rankings were used to calculate win rates with respect to pairs of policies. For harmless dialogue generation, evaluators were instead asked to independently rate each response as harmless or harmful.

For more details on human evaluation, see Appendix I.

## 4 RESULTS

### 4.1 RLAIF VS. RLHF

RLAIF achieves performance gains on par with or better than RLHF on all three tasks (see Figure 1). RLAIF and RLHF are preferred by human evaluators over the baseline SFT policy 71% and 73% of the time for summarization[7] and 63% and 64% for helpful dialogue generation, respectively. The difference in win rates between RLAIF vs. SFT and RLHF vs. SFT are not statistically significant[8]. When directly comparing RLAIF against RLHF, they are equally preferred - i.e. the win rate is not statistically significantly different from 50%[9]. For harmless dialogue generation, RLAIF achieves a harmless rate of 88%, outperforming both RLHF and SFT - 76% and 64%, respectively[10].

We share an example of SFT, RLAIF, and RLHF summaries in Figure 5. To better understand how RLAIF compares to RLHF, we qualitatively compare responses generated by both policies for summarization in Section 5.

As observed in Stiennon et al. (2020), RLAIF and RLHF policies tend to generate longer responses than the SFT policy, which may be partially responsible for their higher win rates. We conduct post-hoc analysis to control for length and find that both RLAIF and RLHF policies still outperform the SFT policy, and by similar margins to one another. See Appendix J for details.

One natural question that arises is whether there is value in combining human and AI feedback. We experimented with combining both types of feedback but did not see an improvement beyond using human feedback alone. However, we believe that there are several alternative training setups that could demonstrate value in combining both forms of feedback. See Appendix K for details.

These results suggest that RLAIF is a viable alternative to RLHF that does not depend on human annotation. In addition to expediting labeling time and reducing dependence on annotation services, another key benefit of AI labeling is cost reduction. We estimate the cost of labeling with a LLM to be more than 10x cheaper than human annotation. See Appendix L for detailed calculations.

---

[6]For summarization, we used the test split of Reddit TL;DR. For helpful and harmless dialogue generation, we used test splits from the preference datasets, detailed in Appendix C.

[7]Additionally, RLAIF and RLHF are preferred over the reference summaries in Reddit TL;DR 79% and 80% of the time, respectively.

[8]For a two-sample t-test, p-value = 0.25 and 0.65 for summarization and helpful dialogue generation, respectively.

[9]The win rate of RLAIF vs. RLHF is 50% for summarization and 52% for helpful dialogue generation.

[10]RLAIF achieves a statistically significant improvement over RLHF and SFT, according to a two-sample t-test.

## 4.2 TOWARDS SELF-IMPROVEMENT

In Section 4.1, the LLM used to label preferences is much larger than the policy LLM (PaLM 2 L vs. PaLM 2 XS). Going one step further, one might wonder if self-improvement is possible - that is, to use the same language model as both the AI labeler and the starting policy. To this end, we set up an experiment on the summarization task where the AI labeler, the RM, and the policy all have the same number of parameters. We then carry out RLAIF as previously described and refer to this setup as "same-size RLAIF".

Human annotators prefer responses from same-size RLAIF 68% of the time over SFT responses. For comparison, our original RLAIF experiment using an AI labeler larger than the policy achieves 71% win rate over SFT. The difference between win rates of same-size "RLAIF vs. SFT" and "RLAIF vs. SFT" is not statistically significant[11]. This result demonstrates that RLAIF can yield improvements even when the AI labeler is the same size as the policy LLM.

In this experiment, the AI labeler and initial policy are not the exact same model. The AI labeler is the instruction-tuned PaLM 2 XS, while the initial policy is PaLM 2 XS fine-tuned on Reddit TL;DR summarization. Additionally, the responses rated by the AI labeler are not generated by other policies created by the original dataset curators. For this reason, this experiment is not strictly "self-improvement". However, we believe that these results show great promise for proper self-improvement.

## 4.3 DIRECT RLAIF

In previous experiments, AI feedback was distilled into a RM. On the summarization task, we experiment with bypassing RM training by using an off-the-shelf LLM to directly provide rewards during RL. Since using a large AI labeler in RL can be costly and slow, we use the smaller instruction-tuned PaLM 2 XS as the off-the-shelf LLM. We refer to this method as "direct RLAIF".

To get direct feedback, we prompt the AI labeler to rate the quality of the current generation between 1 and 10, adding high-level details on the structure of its input and what define a good generation (such as factuality or conciseness for example). We then compute the likelihood of each score token, that is all integers between 1 and 10, normalize the likelihoods to a probability distribution, and calculate a weighted score $s(x|c) = \sum_{i=1}^{10} iP(i|x, c)$, that is then re-normalize to $[-1, 1]$. We give additional details on the prompting method in the Appendix D.

Human annotators prefer responses from direct RLAIF 74% of the time over SFT responses. This result is directly comparable to the same-size RLAIF policy from Section 4.2, which uses the exact same AI labeler and starting policy. Direct RLAIF outperforms same-size RLAIF, which achieves a significantly lower win rate of 68% when compared to SFT. Furthermore, when shown responses side-by-side, raters prefer direct RLAIF over same-size RLAIF 60% of the time. Direct RLAIF outperforms the comparable distilled RLAIF technique, which may be a result of bypassing the distillation step and conveying information directly to the policy.

## 4.4 PROMPTING TECHNIQUES

We experiment with three types of prompting variations - preamble specificity, chain-of-thought reasoning, and few-shot in-context learning (see Table 1). We observe that eliciting chain-of-thought reasoning generally improves AI labeler alignment across all tasks, while the impacts of preamble specificity and in-context learning vary across tasks. The best prompts outperform the base prompts ("Base 0-shot") by +1.9%, +1.3%, and +1.7% for summarization, helpfulness, and harmlessness, respectively.

Preamble specificity consistently improves alignment for summarization (e.g. +1.3% for "Base 0-shot" vs. "Detailed 0-shot"), while giving mixed results helpful and harmless dialogue generation. We hypothesize that summarization benefits more from preamble specificity due to the high complexity of this task. On the other hand, rating helpfulness and harmlessness are more intuitive to grasp, and therefore may benefit less from detailed instructions.

---

[11]The two-sample t-test p-value = 0.07. At alpha = 0.05, this difference is not statistically significant.

Table 1: We observe that eliciting chain-of-thought reasoning tends to improve AI labeler alignment, while few-shot prompting and detailed premables have mixed effects across tasks. H1 refers to helpfulness, H2 to harmlessness.

| Prompt | AI Labeler Alignment | | |
| --- | --- | --- | --- |
| | Summary | H1 | H2 |
| Base 0-shot | 76.1% | 67.8% | 69.4% |
| Base 1-shot | 76.0% | 67.1% | 71.7% |
| Base 2-shot | 75.7% | 66.8% | **72.1%** |
| Base + CoT 0-shot | 77.5% | **69.1%** | 70.6% |
| Detailed 0-shot | 77.4% | 67.6% | 70.1% |
| Detailed 1-shot | 76.2% | 67.6% | 71.5% |
| Detailed 2-shot | 76.3% | 67.3% | 71.6% |
| Detailed 8-shot | 69.8% | – | – |
| Detailed + CoT 0-shot | **78.0%** | 67.8% | 70.1% |
| Detailed + CoT 1-shot | 77.4% | 67.4% | 69.9% |
| Detailed + CoT 2-shot | 76.8% | 67.4% | 69.2% |

Chain-of-thought reasoning improves alignment consistently for summarization. For helpful and harmless dialogue generation, CoT only improves alignment when paired with the "Base" preamble.

Surprisingly, we observe that few-shot in-context learning only improves alignment for harmless dialogue generation[12]. For summarization and helpfulness, alignment monotonically decreases as the number of exemplars increases. We do not believe this decrease is due to low-quality exemplars, which we carefully handpicked high to be representative of each preference task. Furthermore, we conducted 10 trials for "Base 1-shot" on summarization, where we used a different random exemplar for each trial. The maximum AI labeler alignment from these trials was 76.1%, which still did not surpass the "Base 0-shot" alignment. One hypothesis for why exemplars do not help is the summarization and helpful dialogue generation tasks may already be sufficiently well-understood by the powerful AI labeler model, rendering the exemplars useless or even distracting. We also note that in-context learning is still an important research area that is not fully understood (Min et al., 2022; Wang et al., 2022a).

For summarization, we compare against human inter-annotator agreement to get a sense of how well our LLM labeler performs in absolute terms. Stiennon et al. (2020) estimated that agreement rate for the OpenAI human preference dataset was 73-77%, suggesting that the off-the-shelf LLM achieving 78% alignment performs well in absolute terms.

We also conduct experiments with self-consistency. In this technique, multiple chain-of-thought rationales are sampled with temperature $T > 0$, and their resulting preference distributions are averaged together. We find that self-consistency strictly degrades AI labeler alignment (see Appendix M).

We expect that higher AI labeler alignment in theory should lead to improvements in RLAIF policies. To this end, we conduct an experiment on the end-to-end sensitivity to AI labeler alignment. We train two RLAIF policies that only differed in the alignment scores of AI labels. We observe that the policy trained with more aligned AI labels achieves a significantly higher win rate. However, this study only compares two policies, and rigorous experimentation is required to draw certain conclusions. See Appendix N for details.

## 4.5 SIZE OF LLM LABELER

Table 2: AI labeler alignment increases as the size of the LLM labeler increases.

| Model Size | AI Labeler Alignment |
| --- | --- |
| PaLM 2 XS | 62.7% |
| PaLM 2 S | 73.8% |
| **PaLM 2 L** | **78.0%** |

---

[12]We verified that all examples used in this experiment fit within our AI labeler's context length.

Large model sizes are not widely accessible and can be slow and expensive to run. On the task of summarization, we experiment with labeling preferences with varying LLM sizes and observe a strong relationship between size and alignment. Alignment decreases -4.2% when moving from PaLM 2 Large (L) to PaLM 2 Small (S), and it decreases another -11.1% when moving down to PaLM 2 XS - a trend consistent with scaling behaviors observed in other work (Kaplan et al., 2020). In addition to being less powerful models, another contributing factor to the decline in performance could be the increase in position bias in smaller LLMs (see Appendix B).

On the other end of this trend, these results also suggest that scaling up the AI labeler size may produce even higher quality preference labels. Since the AI labeler is only used to generate preference examples once and is not called during RL, using an even larger AI labeler is not necessarily prohibitively expensive.

## 5 QUALITATIVE OBSERVATIONS

To better understand how RLAIF compares to RLHF, we inspected responses generated by both policies for the summarization task. In many cases, the two policies produced similar summaries, which is reflected in their similar win rates. However, we identified two patterns where they occasionally diverged.

The first pattern we observed is that sometimes RLAIF hallucinates less than RLHF. The hallucinations in RLHF summaries were plausible but inconsistent with the original text. For instance, in Example #1 of Table 22, the RLHF summary states that the author is 20 years old, but this is not mentioned or implied by the original text. The second pattern we observed is that RLAIF sometimes produced less coherent or grammatical summaries than RLHF. For instance, in Example #1 of Table 23, the RLAIF summary produces run-on sentences.

More systematic analysis is required to identify if these patterns exist at scale. We leave this to future work.

## 6 RELATED WORK

LLMs have shown impressive performance over a wide range of NLP tasks (Brown et al., 2020; Thoppilan et al., 2022; Chowdhery et al., 2022; Google et al., 2023; OpenAI, 2023a). For several of these tasks, RL has emerged as an effective optimization technique. While initial applications of RL on tasks such as translation (Wu et al., 2016; 2018) and summarization (Gao et al., 2019; Wu and Hu, 2018) used automatic evaluation metrics as rewards, such simplified formulations of rewards did not fully align with human notions of quality.

Reinforcement learning from human feedback (Christiano et al., 2017) has been used as a technique to directly align LLMs with human preferences (Ziegler et al., 2019) through training a reward model on pairwise comparisons of natural language responses. It has been successfully applied for summarization (Stiennon et al., 2020), generalized instruction following (Ouyang et al., 2022; Lai et al., 2023), dialogue (Gilardi et al., 2023; Manyika, 2023; Glaese et al., 2022; Bai et al., 2022a) and question answering (Nakano et al., 2021).

LLMs have also been extensively used for data generation (Wang et al., 2021b; Meng et al., 2023), augmentation (Feng et al., 2021) and in self-training setups (Wang et al., 2022b; Madaan et al., 2023). Bai et al. (2022b) introduced the idea of RLAIF, which used LLM labeled preferences in conjunction with human labeled preferences to jointly optimize for the two conflicting objectives of helpfulness and harmlessness. Recent works have also explored related techniques for generating rewards from LLMs (Roit et al., 2023; Kwon et al., 2022; Yang et al., 2023). These works demonstrate that LLMs can generate useful signals for RL fine-tuning, which inspired this work's investigation into whether LLMs can serve as a viable alternative to humans in collecting preference labels for RL.

## 7 CONCLUSION

In this work, we show that RLAIF achieves comparable improvements to RLHF. Our experiments show that RLAIF greatly improves upon a SFT baseline, and the margin of improvement is on par

with that of RLHF. Furthermore, in head-to-head comparisons, RLAIF and RLHF are preferred at similar rates by humans. Additionally, we show that RLAIF is effective even when the LLM labeler is the same size as the policy, and directly prompting the LLM labeler for rewards at RL can outperform the canonical RLAIF setup that distills preferences into a separate RM. Finally, we study the impact of AI labeling techniques on alignment to human preferences.

While this work highlights the potential of RLAIF, there remain many fascinating open questions, such as whether conducting RLAIF iteratively can bring additional gains (i.e. use the RLAIF policy to generate new response pairs, conduct RLAIF, and repeat), how RLAIF can be adapted to a model-based RL setting where both human and assistant are modeled by LLMs, and how AI feedback can be leveraged for more specific credit assignment. We leave these questions for future work.

## ETHICS

In conducting our research, we have adhered to strict ethical principles to ensure the integrity and responsibility of our work. Prior to participating in the preference rating task, all human raters provided informed consent. Additionally, we compensated the human participants fairly for their time and contributions.

A primary ethical consideration concerns the utilization of AI-generated feedback as a source for model alignment. There exists a potential risk of inheriting biases from the pre-trained off-the-shelf LLM into the generated labels. This in turn may result in models which amplify the biases from pre-trained data. We must exercise extreme caution especially when deploying these models in high-stakes domains such as medicine, law, and employment, where they have the potential to significantly impact human lives in adverse ways.

Furthermore, reducing the barriers to aligning LLMs also carries the risk of facilitating their misuse for malicious purposes. For instance, they could be employed to generate convincing misinformation or produce hateful and abusive content.

## REPRODUCIBILITY

To promote reproducibility of our work, we list the open-source datasets used in Section 3.1, the LLM labeling details in Section D, model training hyper-parameters in Appendix G, RL algorithms in Appendix F, and prompts used in Appendix Tables (e.g. Tables 15 and 16). PaLM 2 models are available through Google Cloud's Vertex API, and the experiments in this work may also be repeated with other publicly available LLMs.

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

## A  RLHF PRELIMINARIES

We review the RLHF pipeline introduced in Stiennon et al. (2020); Ouyang et al. (2022), which consists of 3 phases: supervised fine-tuning, reward model training, and reinforcement learning-based fine-tuning.

### A.1  SUPERVISED FINE-TUNING

A pre-trained LLM is fine-tuned on a high quality labeled dataset for a downstream task (e.g. given an input document, generate a summary) using token-level supervision to produce a supervised fine-tuned (SFT) model $\pi^{SFT}$.

### A.2  REWARD MODELING

Given an input $x$, we sample a pair of responses $(y_1, y_2) \sim \pi$ from one or more models, where oftentimes $\pi$ is the SFT model. The input and responses are sent to human annotators to rate which response is better according to some criteria. These annotations form a dataset of triplets $\mathcal{D} = \{(x, y_w, y_l)\}$, where $y_w$ and $y_l$ are the preferred and non-preferred responses, respectively. A reward model (RM) $r_\phi$ is trained by minimizing the following loss:

$$\mathcal{L}_r(\phi) = \underset{(x,y_w,y_l)\sim\mathcal{D}}{-\mathbb{E}}\left[\log\sigma\big(r_\phi(x, y_w) - r_\phi(x, y_l)\big)\right],$$

where $\sigma$ is the sigmoid function.

### A.3  REINFORCEMENT LEARNING

A policy $\pi_\theta^{RL}$ is initialized from the SFT model weights and then optimized with reinforcement learning to maximize the reward given by the RM, which serves as a proxy for human preferences. Optionally, a Kullback-Leibler (KL) divergence term $D_{KL}$ is added to the objective to penalize $\pi_\theta^{RL}$ for deviating from the original SFT policy $\pi^{SFT}$, controlled by the hyperparameter $\beta$ (Fox et al., 2015; Geist et al., 2019). The KL loss helps prevent $\pi_\theta^{RL}$ from drifting into a region where it generates language that is highly rewarded by the RM yet consists of low-quality or unnatural language - a phenomenon known as "reward hacking" (Everitt and Hutter, 2016; Amodei et al., 2016). The optimization objective is described by the equation below:

$$J(\theta) = \underset{y\sim\pi_\theta(\cdot|x)}{\mathbb{E}}\Big[(1-\beta)r_\phi(y|x) \\ - \beta D_{KL}\big(\pi_\theta^{RL}(y|x) \,||\, \pi^{SFT}(y|x)\big)\Big].$$

## B  POSITION BIAS IN LLM LABELERS

Table 3: Position bias is more prevalent in smaller model sizes, measured by the percentage of examples where the LLM prefers the same position even after swapping the order of candidates ("% Same Position Preferred"). Analysis is conducted using the "Detailed + CoT 0-shot" prompt.

| Model Size | % Same Position Preferred |
|---|---|
| PaLM 2 L | 18% |
| PaLM 2 S | 21% |
| PaLM 2 XS | 56% |

Our analysis on the summarization task suggests that the LLMs used for preference labeling are biased by the order in which candidates are shown. For each example in our AI labeling evaluation

set, we query the LLM preferences for the pair of candidates, swap the order in which candidates are presented, and then query the LLM preferences again.

We consider a LLM to be *more biased* if it prefers the same position on both the original and reversed inferences. For example, let candidates A and B be in positions 1 and 2 for the first inference and in positions 2 and 1 for the second, respectively. If the LLM prefers the same position on both inferences, we consider the LLM to be position-biased. We measure position bias by computing *"% Same Position Preferred"* - the percentage of inference pairs where this occurs, and a higher metric value indicates a more biased LLM.

We find that PaLM 2 L, S, and XS prefer the same position 18%, 21%, and 56% of the time, respectively (see Table 3), suggesting that position bias is inversely correlated with model size. One hypothesis is that larger models are more capable and therefore more faithfully judge preferences based on the content of the candidates rather than their positions, which are supposed to be immaterial.

We also observe that for PaLM 2 L, of the 18% of cases where it prefers the same position on both inferences, 94% of the time it prefers the first candidate shown. On the other hand, PaLM 2 S and XS show affinity for the second candidate shown, preferring it 91% and 99% of the time, respectively, when the same position is preferred on both inferences. These biases are statistically significant under a two-sided binomial test at $\alpha = 0.05$.

## C   DATASET DETAILS

For summarization, we use the filtered Reddit TL;DR dataset (Stiennon et al., 2020), containing posts from Reddit[13] that have been filtered to ensure high quality. The dataset contains 123k posts, and ~5% is held out as a validation set.

Additionally, we use OpenAI's human preference dataset created from the filtered TL;DR dataset. For a given post, two candidate summaries were generated from different policies, and human labelers were asked to rate which summary they preferred. The total dataset comprises 92k pairwise comparisons.

For helpful and harmless dialogue generation, we use Anthropic's Helpful and Harmless preference datasets[14] (Bai et al., 2022a), which consists of conversation history between a human and an AI assistant and a preferred and non-preferred response from the AI assistant. Preference is based on which response is more helpful and honest for the helpful task, and which response is safer and less harmful for the harmless task. Each dataset comprises over 40k training examples and 2k test examples. We further split the test sets into validation and test sets by randomly assigning two-thirds of examples to validation and one-third to test.

## D   LLM LABELING DETAILS

For LLM labeling, we set a maximum input context length of 4096 tokens. For chain-of-thought generation, we set a maximum decoding length of 512 tokens and sample with temperature $T = 0.0$ (i.e. greedy decoding). For self-consistency experiments, we use temperatures varying from $T = 0.3$ to $T = 1.0$ with top-K sampling (Fan et al., 2018), where $K = 40$.

In Section 4.3, we used the AI labeler to compute a score that we leverage as direct reward in the RLAIF procedure. We use the following prompt: *"You are an expert summary rater. Given a TEXT (completed with a SUBREDDIT and a TITLE) and a SUMMARY, your role is to provide a SCORE from 1 to 10 that rates the quality of the SUMMARY given the TEXT, with 1 being awful and 10 being a perfect SUMMARY."*, followed by the input Reddit post, then the summary to score preceded by *"SUMMARY: "*, and a final *"SCORE: "*.

PaLM 2 models are publicly available through Google Cloud's Vertex AI[15], though we cannot guarantee full reproducibility as the models accessible through Google Cloud are subject to change.

---

[13] www.reddit.com

[14] We use the `helpful-base` and `harmless-base` datasets from https://huggingface.co/datasets/Anthropic/hh-rlhf.

[15] https://cloud.google.com/vertex-ai/docs/generative-ai/learn/models

## E  POST-RL RESPONSE FORMATTING

Post-RL (both RLHF and RLAIF) models have a tendency to "hack" the reward by adding superfluous symbols like periods or spaces at the end of the response. As these extra tokens do not have any meaningful content, we remove trailing superfluous spaces or periods without altering the content. This makes human judgement easier and fairer, as the judgement is not biased by formatting unrelated to the content of the response.

## F  REINFORCE FOR LANGUAGE MODELS

Consider a deterministic, finite-horizon MDP $M = (\mathcal{X}, \mathcal{A}, R, P, \gamma)$ (Howard, 1960). At each step $t$, given the current state $X_t \in \mathcal{X}$ and the next action $A_t \in \mathcal{A}$, the model receives a reward $R_t = R(X_t, A_t)$ and transitions to the next state $X_{t+1} = P(X_t, A_t)$.

In the context of language models, $X_t$ is the concatenation of the input text and all text the policy has generated up to time $t$. Action $A_t$ is the token decoded at time $t$ by the stochastic policy $\pi_\theta(\cdot|X_t)$ from the considered vocabulary, where $\theta$ represents the policy parameters. Finally, the reward $R_t$ is given by the RM. The RM is only evaluated when the language model response has been fully generated; therefore all rewards prior to the last token are set to be 0, while the reward corresponding to the final token is set to be $R_T$.

The cumulative sum of rewards received when following the policy $\pi_\theta$ from a time-step $t$ is called the return. Generally, it is defined as $Z_t = \sum_{s=t}^{T} \gamma^{s-t} R_s$. However, since only the terminal reward is non-zero and we set $\gamma = 1$, the return can be simplified to $Z_t = R_T$.

Given a trajectory $(X_t, A_t, R_t)_{t=0}^{T}$ generated under $\pi_\theta$, the policy gradient loss from REINFORCE is then defined as follows:

$$\mathcal{L}_{\text{PG}}(\theta) = -\sum_t \log \pi_\theta(A_t|X_t)\overline{\Big(Z_t - V_\psi^\pi(X_t)\Big)},$$

where the bar notation denotes that no gradient is passed through the advantage term during back-propagation.

The baseline value function $V_\psi^\pi(x)$ estimates the return-to-go $Z_t$ when following the policy $\pi_\theta$ and is parameterized by $\psi$ (Williams, 1992; Sutton et al., 1999). It is trained with the following loss:

$$\mathcal{L}_V(\psi) = \sum_t (Z_t - V_\psi^\pi(X_t))^2.$$

In practice, given that we optimize for the regularized objective in Sec. A.3, we incorporate the KL divergence in the policy gradient loss, as commonly done in the literature (Jaques et al., 2017).

## G  MODEL TRAINING DETAILS

Model training consists of 3 phases, supervised fine-tuning, reward model training and reinforcement learning. We alter settings of model training as needed for each of the 3 tasks.

We train SFT models for the summarization task on the Reddit TL;DR dataset, with a batch size of 128 for a single epoch. We use the Adafactor (Shazeer and Stern, 2018) optimizer with a learning rate of $10^{-5}$, and we set maximum input and output lengths of 1024 and 128 tokens, respectively. For helpful and harmless dialogue generation tasks, we treat an instruction-tuned version of PaLM 2 XS as the SFT model.

We train RMs for all tasks until the training loss and accuracy curves plateau, which happens in 2-3 epochs. We use the Adafactor optimizer with a learning rate of $10^{-5}$. Batch size is 128 for summarization RMs and 32 for RMs of other tasks. We train all our RMs with maximum input length of 1152 tokens, comprising of 1024 tokens for the context and 128 tokens for the response. We report the pairwise accuracies of the RMs in Table 4.

For summarization, we initialize the AI feedback RM from the SFT model (i.e. PaLM 2 XS fine-tuned on Reddit TL;DR) and the human feedback RM from PaLM 2 XS. We experimented with initializing

the human feedback RM from the SFT model but found that it resulted in lower pairwise accuracy on the held out set of human preferences (see Table 5). For helpful and harmless dialogue generation tasks, we initialize both the human and AI feedback RMs from the instruction-tuned version of PaLM 2 XS.

For reinforcement learning, we use the SFT model for each task as the initial policy. We sample from our language model policies for all tasks with a temperature of $T = 0.9$ to encourage exploration. We train with a batch size of 128 and learning rate of $10^{-5}$ for 8 epochs, resulting in ∼1 million episodes. We set $\beta = 0.05$ for the KL divergence loss.

To select a final checkpoint for each RL policy, we first selected 4 candidate checkpoints from RL training that scored high rewards on validation prompts. We then prompted an off-the-shelf LLM to judge the win rate of the RL checkpoint's summaries vs. the SFT policy's summaries. We also conducted manual inspection of a dozen examples. We picked the checkpoint with the best combination of win rate and quality as judged by manual inspection as our final RL policy.

## H  REWARD MODEL ACCURACY

Table 4: Pairwise accuracies of human feedback and AI feedback reward models across all tasks. Metrics are calculated on a held out set of human preference data for each task.

| Tasks | Human Feedback | AI Feedback |
|---|---|---|
| Summarization | 79.3% | 74.2% |
| Helpful Dialogue | 76.0% | 67.8% |
| Harmless Dialogue | 72.1% | 69.7% |

Table 5: Results of initializing the summarization RMs on PaLM 2 XS vs. the SFT model.

| Initialization | Human Feedback | AI Feedback |
|---|---|---|
| PaLM 2 XS | **79.3%** | 73.0% |
| SFT | 78.7% | **74.2%** |

Table 6: Accuracy values for variants of RMs trained on AI labels for the task of summarization.

| RM Variant | AI Feedback |
|---|---|
| Trained on "Base 0-shot" labels | 77.9% |

*Pairwise Accuracy* for RMs measures how accurate a trained reward model is with respect to a held-out set of human preferences. Given an input context and pair of candidate responses, the *Pairwise Accuracy* is 1 if the RM scores the preferred candidate higher than the non-preferred candidate, according to the human label. Otherwise the value is 0. This quantity is averaged over multiple examples to obtain the total pairwise accuracy of the RM.

We report RM pairwise accuracy on a held out set of human preferences for all tasks in Table 4. For summarization, we also report RM pairwise accuracy when initializing on different checkpoints in Table 5 and on other RM variants in Table 6.

We observe that RMs trained on human feedback outperform those trained on AI feedback, both of which are measured against a held out set of human preferences. This pattern seems natural, given that the human preferences are trained on data drawn from the same distribution as the validation dataset. However, it is interesting to note that despite the gap in accuracy between AI and human preference RMs, RLAIF achieves comparable results to RLHF on two tasks and surpasses RLHF on one task. Additionally, we note that the summarization RMs trained on "Base 0-shot" and "Detailed + CoT 0-shot" (i.e. the default prompting technique) achieve accuracies of 77.9% and 74.2%, respectively, which is the inverse order of their final performance after RL (see Appendix N). These gaps in RM

accuracy suggest that RM accuracy, while correlated with RM usefulness, may not be a perfect reflection of RM effectiveness in RLHF and RLAIF. Ultimately, we believe that the usefulness of RMs is assessed through conducting RLHF and RLAIF and evaluating the final policies through human evaluation.

## I  HUMAN EVALUATION DETAILS

To conduct human evaluation, in total we created ∼2k unique rating instances. Each instance comprised a single context and three distinct model responses (e.g. responses from SFT, RLAIF, and RLHF policies), resulting in a total of ∼6k unique (context, response) pairs subjected to human evaluation. Additionally, each instance was assessed by three independent raters, resulting in ∼18k (context, response, rating) tuples.

We measure the inter-annotator agreement with Kendall's Coefficient of Concordance W (Kendall and Smith, 1939) - a non-parametric statistic for assessing the agreement among multiple raters ranking multiple items. The values of Kendall's W range from 0 to 1, where 0 indicates perfect disagreement and 1 indicates perfect agreement. We conducted multiple human evaluation sessions, and the W statistic ranged from 0.6-0.7, indicating a reasonable level of agreement.

## J  CONTROLLING FOR RESPONSE LENGTH

Our RLAIF and RLHF policies generate responses that differ in length from our baselines such as the SFT policy or human generations. For example, in the summarization task, the summaries produced by the RLAIF, RLHF, and SFT policies sent to human evaluation have an average character-length of 164, 161, and 132, respectively. For all experiments presented in this paper, we conduct post-hoc analysis to estimate the win rates of RLAIF and RLHF vs. SFT after controlling for length.

We take an approach similar to Stiennon et al. (2020). For each of our RL policies, we train a logistic regression model where the input is the ratio of the RL summary length to the SFT summary length (in characters) and the target is a binary label indicating whether RL was preferred to SFT. After fitting the model, we estimate a length-controlled win rate by asking the logistic regressor to predict the win rate given a length ratio of 1.0, which represents the scenario where both the RL and SFT summaries are of equal length.

After controlling for length, in the summarization task, our estimated win rates for RLAIF and RLHF vs. SFT are 59% and 61%, respectively (see Table 7). Both RL policies continue to outperform the SFT policy by a similar margin, supporting our initial conclusion that RLAIF is comparable to RLHF.

We reach similar conclusions for the helpful dialogue generation task (Table 8). Similarly, results hold for the experiments looking at the end-to-end sensitivity to AI labeler alignment N (Table 10), also when combining human and AI feedback K (Table 11) and finally also for the experiments towards self-improvement 4.2 (Table 12).

We note that for the harmless dialogue generation task, the setup is slightly different. Indeed, as humans provided binary feedback (i.e. harmful or harmless), we compute the harmless rate instead of the win rate when getting ordering of the outputs of the different models from humans. Here we used the average generation length from the SFT model as reference to compute, as done before, the length-controlled harmless rate for RLHF and RLAIF (Table 9).

We note that this post-hoc method of controlling for length is imperfect, as it assumes the logistic regression model can accurately learn the relationship between summary length and human preference. A more principled approach would be to have all policies generate summaries of similar length (e.g. by encouraging policies to generate summaries of a fixed length during optimization).

## K  COMBINING HUMAN AND AI FEEDBACK

We investigate the effectiveness of combining human feedback and AI feedback. We call this approach RLHF + RLAIF, and compare it against RLHF. We conduct this preliminary experiment on the TL;DR summarization task.

Table 7: Length-controlled win rate for the summarization task.

| Models | Length uncorrected | Length corrected |
|---|---|---|
| RLAIF vs SFT | 71% | 59% |
| RLHF vs SFT | 73% | 61% |
| RLAIF vs RLHF | 50% | 47% |
| RLAIF vs Reference | 79% | 74% |
| RLHF vs Reference | 80% | 76% |

Table 8: Length-controlled win rate for the helpful dialogue generation task.

| Models | Length uncorrected | Length corrected |
|---|---|---|
| RLAIF vs SFT | 63% | 61% |
| RLHF vs SFT | 64% | 61% |
| RLAIF vs RLHF | 52% | 50% |

To perform RLHF + RLAIF, we start with a model trained via RLHF and a model trained via SFT, and collect responses from both at a high temperature of 1.0 to increase diversity. We then use our AI labeler to generate AI feedback and collect preferences for these responses. We now train a new reward model using both Human and AI preference data, and perform RL fine-tuning with it.

To evaluate the new RLHF + RLAIF model, we show human evaluators SFT responses, RLHF responses and RLHF + RLAIF responses. We see that combining 2 sources of feedback performs similar to training only with human feedback, i.e. empirically it brings no incremental advantage. Human annotators prefer responses from RLHF 74% of the time over SFT responses while they prefer responses from RLHF + RLAIF 71% of the time over SFT responses. The difference in win-rate is not statistically significant.[16]. When shown responses side-by-side, raters prefer them equally. RLHF + RLAIF has a win-rate of 48% but not statistically different from 50%.

Our experiment did not show positive results from combining RLAIF and RLHF. However, we believe that there are many alternative experimental setups which could demonstrate utility in combining AI and human feedback. One setup could involve first conducting RLAIF, then collecting generations and human preferences using the RLAIF policy for RLHF. This curriculum learning approach treats RLAIF as a "warm-up" policy, which could then be refined with RLHF. Another setup could involve collecting much more AI feedback than human feedback, since it is much less expensive to collect. We leave this exploration to future work.

## L    COST OF LLM VS. HUMAN LABELING

Using LLMs as data annotators can be much less costly than hiring human annotators (Wang et al., 2021a). We estimate AI preference labeling to be over 10x less costly than human preference labeling using the calculations below.

At the time of writing, GPT-4 charged $0.03 USD and $0.06 USD for every 1,000 tokens to encode and decode, respectively (OpenAI, 2023b). For labeling TL;DR preferences with a LLM, our average token lengths were as follows:

1. *Input prompt length* - 830 tokens (using the "Detailed + CoT 0-shot" prompt (see Table 16)
2. *Generated chain-of-thought rationale* - 61 tokens
3. *"1" and "2" decoded for preference distribution* - 2 tokens

Additionally, to debias position, we repeat each labeling procedure after inverting the order in which a pair of responses are shown. Our estimated AI labeling cost per example is $0.06 USD[17].

---

[16]We conduct a two-sample t-test and find that, p-value=0.15. So we can reject the null hypothesis here

[17]2 inferences * (830 encoder tokens * $0.03 / 1,000 tokens + (61 + 2) decoder tokens * $0.06 / 1,000 tokens) = $0.057 ∼ = $0.06

Table 9: Length-controlled harmless rate for the harmless dialogue generation task. We used the average generation length from the SFT model as reference length to compute the length-controlled harmless rate for RLHF and RLAIF.

| Models | Length uncorrected | Length corrected |
|--------|--------------------|------------------|
| SFT    | 64%                | 64%              |
| RLHF   | 76%                | 78%              |
| RLAIF  | 88%                | 91%              |

Table 10: Length-controlled win rate for experiments looking at end-to-end sensitivity to the AI labeler alignment. Base RLAIF and Detailed RLAIF respectively correspond to Base 0-shot RLAIF and Detailed CoT 0-shot RLAIF described in N.

| Models | Length uncorrected | Length corrected |
|--------|--------------------|------------------|
| Base RLAIF vs SFT | 63% | 59% |
| Detailed RLAIF vs SFT | 67% | 63% |
| Base RLAIF vs Detailed RLAIF | 41% | 45% |

For human annotation, Google Cloud's AI Platform Data Labeling Service charged approximately $0.11 USD / 50 words for classification tasks at the time of writing[18] (Google, 2023). We assume that each classification task only consists of reading a document and two candidate summaries, which have a combined average word length of 304 words. We estimate the human labeling cost per example to be $0.67 USD (304 words * $0.11 / 50 words).

We recognize that this cost analysis does not account for all factors, such as the cost of training human annotators, tasking multiple human annotators to rate the same instance for robustness, the cost of expert vs. crowd-sourced annotators, or the cost of setting up LLM labeling.

## M  SELF-CONSISTENCY

For chain-of-thought prompts, we also experiment with self-consistency (Wang et al., 2022b) - a technique to improve upon chain-of-thought reasoning. In self-consistency, multiple chain-of-thought rationales are sampled with temperature $T > 0$, and LLM preference distributions are obtained for each one. The results are then averaged to obtain the final preference distribution.

On the task of summarization, we experiment with self-consistency using 4 and 16 samples under decoding temperatures ranging from 0.3 to 1.0 (see Figure 13)[19]. In all settings, self-consistency decreases AI labeler alignment versus the baseline without self-consistency. Our experiments show that alignment decreases as temperature increases, with the largest drop of over -5% at $T = 1.0$. In our experiments, using 4 vs. 16 self-consistency samples does not impact AI labeler alignment.

Manually inspecting chain-of-thought rationales did not reveal any common patterns for why self-consistency might degrade alignment (examples in Table 19). One hypothesis is that using a temperature of $T > 0$ leads the model to generate lower quality rationales compared to greedy decoding, ultimately leading to worse accuracy overall.

---

[18]Google Cloud charges between $90 and $129 per 1,000 units, where each unit is 50 words for a classification task. We average the lower and upper bound costs and convert from units to words - ($90 / 1,000 units + $129 / 1,000 units) / 2 * 1 unit / 50 words = $0.1095 USD / 50 words

[19]Results of using 4 samples are not shown because they only differ from the 16-sample results by $\pm 0.4\%$.

Table 11: Length-controlled win rate for experiments combining human and AI feedback.

| Models | Length uncorrected | Length corrected |
|---|---|---|
| RLHF + RLAIF vs SFT | 71% | 61% |
| RLHF vs SFT | 74% | 67% |
| RLHF + RLAIF vs RLHF | 48% | 46% |

Table 12: Length-controlled win rate for experiments towards self-improvement.

| Models | Length uncorrected | Length corrected |
|---|---|---|
| Direct RLAIF vs SFT | 74% | 65% |
| Distilled RLAIF vs SFT | 68% | 59% |
| Direct RLAIF vs Distilled RLAIF | 60% | 56% |

Table 13: Sampling several chain-of-thought rationales with $T > 0$ results in lower alignment with human preferences. Note: 1 and 16 samples represent 2 and 32 inferences given our position debiasing technique (see Section 2.1.1).

| Self-Consistency | AI Labeler Alignment |
|---|---|
| **1 sample, T=0.0** | **78.0%** |
| 16 samples, T=0.3 | 76.2% |
| 16 samples, T=0.5 | 75.1% |
| 16 samples, T=0.7 | 74.0% |
| 16 samples, T=1.0 | 72.8% |

# N    END-TO-END SENSITIVITY TO AI LABELER ALIGNMENT

We assess the end-to-end sensitivity of the final RL policies to AI labeler alignment on the task of summarization. Since human judgement is subjective and prone to noise, we test whether higher "human alignment" leads to improved downstream performance. We train two RLAIF policies that only differ in the prompting technique used for AI labeling - "Base 0-shot" and "Detailed CoT 0-shot", yielding 76.1% and 78.0% AI labeler alignment, respectively.

When compared head-to-head, human evaluators prefer responses from the policy derived from the more aligned prompting technique 59% of the time[20]. This result suggests that small gains in AI labeler alignment may lead to improvements in the final RL policies. However, we acknowledge that this study is limited, and further experiments are required to draw generalizable conclusions.

We report the accuracy of both RMs in Appendix H.

---

[20]Result is statistically significantly different from 50%.

Table 14: An example of a prompt fed to an off-the-shelf LLM to generate AI preference labels. "{text}", "{summary1}", and "{summary2}" are populated with unlabeled examples, and a preference distribution is obtained by computing the softmax of the log-probabilities of generating the tokens "1" vs. "2".

| Preamble | A good summary is a shorter piece of text that has the essence of the original. ... Given a piece of text and two of its possible summaries, output 1 or 2 to indicate which summary best adheres to coherence, accuracy, coverage, and overall quality as defined above. |
|---|---|
| Exemplar | >>>>>>>> Example >>>>>>>> 

 Text - We were best friends over 4 years ... 
 Summary 1 - Broke up with best friend, should I wish her a happy birthday... And what do you think of no contact? 
 Summary 2 - should I wish my ex happy birthday, I broke no contact, I'm trying to be more patient, I'm too needy, and I don't want her to think I'll keep being that guy. 

 Preferred Summary=1 

 >>>>>>>> Follow the instructions and the example(s) above >>>>>>>> |
| Sample to Annotate | Text - {**text**} 
 Summary 1 - {**summary1**} 
 Summary 2 - {**summary2**} |
| Ending | Preferred Summary= |

Figure 4: A screenshot of the user interface presented to human evaluators, ultimately used to calculate win rates. Raters are shown a context and asked to rank the quality of candidate responses.

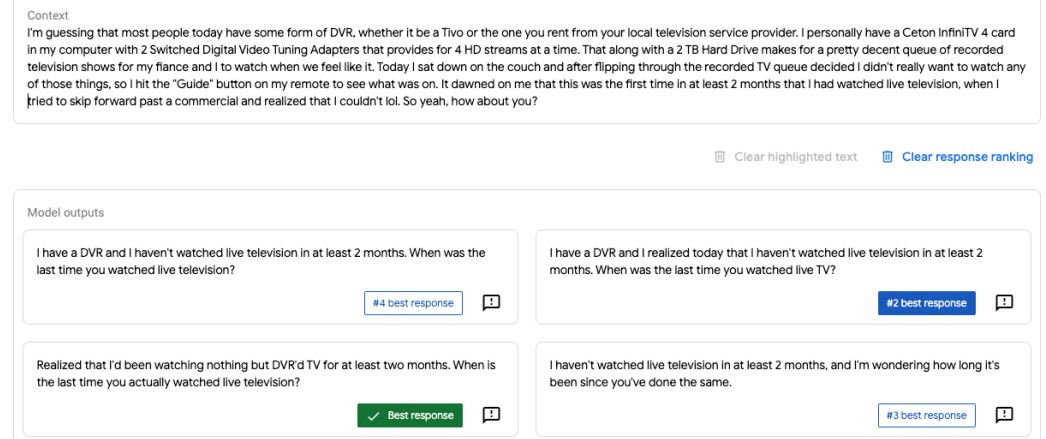

Table 15: The "Base" and "Detailed" preambles given to the LLM labeler to obtain preference labels for the summarization task.

| | |
|---|---|
| "Base" preamble | You are an expert summary rater. Given a piece of text and two of its possible summaries, output 1 or 2 to indicate which summary is better. |
| "Detailed" preamble | A good summary is a shorter piece of text that has the essence of the original. It tries to accomplish the same purpose and conveys the key information from the original post. Below we define four evaluation axes for summary quality: coherence, accuracy, coverage, and overall quality.

Coherence: This axis answers the question \how coherent is the summary on its own?" A summary is coherent if it's easy to understand when read on its own and free of English errors. A summary is not coherent if it's difficult to understand what the summary is trying to say. Generally, it's more important that the summary is understandable than it being free of grammar errors.

Accuracy: This axis answers the question \does the factual information in the summary accurately match the post?" A summary is accurate if it doesn't say things that aren't in the article, it doesn't mix up people, and generally is not misleading.

Coverage: This axis answers the question \how well does the summary cover the important information in the post?" A summary has good coverage if it mentions the main information from the post that's important to understand the situation described in the post. A summary has poor coverage if someone reading only the summary would be missing several important pieces of information about the situation in the post. A summary with good coverage should also match the purpose of the original post (e.g. to ask for advice).

Overall quality: This axis answers the question \how good is the summary overall at representing the post?" This can encompass all of the above axes of quality, as well as others you feel are important. If it's hard to find ways to make the summary better, the overall quality is good. If there are lots of different ways the summary can be made better, the overall quality is bad.

You are an expert summary rater. Given a piece of text and two of its possible summaries, output 1 or 2 to indicate which summary best adheres to coherence, accuracy, coverage, and overall quality as defined above. |

Table 16: The template used for the "Detailed + CoT 0-shot" prompt for summarization. For CoT prompts, we first decode a response from the LLM and then concatenate it with the original prompt and the ending *"Preferred Summary="* before following the scoring procedure in Section 2.1 to obtain a preference distribution.

| | |
|---|---|
| Preamble | A good summary is a shorter piece of text that has the essence of the original. It tries to accomplish the same purpose and conveys the key information from the original post. Below we define four evaluation axes for summary quality: coherence, accuracy, coverage, and overall quality.

Coherence: This axis answers the question \how coherent is the summary on its own?" A summary is coherent if it's easy to understand when read on its own and free of English errors. A summary is not coherent if it's difficult to understand what the summary is trying to say. Generally, it's more important that the summary is understandable than it being free of grammar errors.

Accuracy: This axis answers the question \does the factual information in the summary accurately match the post?" A summary is accurate if it doesn't say things that aren't in the article, it doesn't mix up people, and generally is not misleading.

Coverage: This axis answers the question \how well does the summary cover the important information in the post?" A summary has good coverage if it mentions the main information from the post that's important to understand the situation described in the post. A summary has poor coverage if someone reading only the summary would be missing several important pieces of information about the situation in the post. A summary with good coverage should also match the purpose of the original post (e.g. to ask for advice).

Overall quality: This axis answers the question \how good is the summary overall at representing the post?" This can encompass all of the above axes of quality, as well as others you feel are important. If it's hard to find ways to make the summary better, the overall quality is good. If there are lots of different ways the summary can be made better, the overall quality is bad.

You are an expert summary rater. Given a piece of text and two of its possible summaries, explain which summary best adheres to coherence, accuracy, coverage, and overall quality as defined above. |
| Sample to Annotate | Text – {text}
Summary 1 – {summary1}
Summary 2 – {summary2} |
| Ending | Consider the coherence, accuracy, coverage, and overall quality of each summary and explain which one is better.

Rationale: |

Table 17: The template used for the "Detailed + CoT 1-shot" prompt for summarization, with some text removed for brevity.

| | |
|---|---|
| Preamble | A good summary is a shorter piece of text that has the essence of the original. ... Given a piece of text and two of its possible summaries, explain which summary best adheres to coherence, accuracy, coverage, and overall quality as defined above. |
| Exemplar | >>>>>>>> Example >>>>>>>>

Text – We were best friends over 4 years ...
Summary 1 – Broke up with best friend, should I wish her a happy birthday... And what do you think of no contact?
Summary 2 – should I wish my ex happy birthday, I broke no contact, I'm trying to be more patient, I'm too needy, and I don't want her to think I'll keep being that guy.

Thoughts on Summary 1 –
Coherence – 7. Rationale: The summary is generally understandable, though it could be written with better grammar.
Accuracy – 9. Rationale: The summary doesn't say things that aren't in the original text, and isn't misleading.
Coverage – 6. Rationale: The summary covers most of the important information in the post and conveys the gist of the original text. However, it places more emphasis on ``no contact'' and could have mentioned the smothering/neediness to be more complete.
Overall Quality – 7. Rationale: The summary represents the post fairly well with only minor areas where it could be improved.

Thoughts on Summary 2 –
Coherence – 3. Rationale: The summary is long-winded and has several grammatical errors.
Accuracy – 4. Rationale: The summary mentions that the author broke no contact, but this is incorrect. Otherwise, it is accurate.
Coverage – 8. Rationale: The summary covers the key points in the original text.
Overall Quality – 4. Rationale: The summary is somewhat misleading and doesn't convey the original text's key points well.

Preferred Summary=1

>>>>>>>> Follow the instructions and the example(s) above >>>>>>>> |
| Sample to Annotate | Text – {text}
Summary 1 – {summary1}
Summary 2 – {summary2} |
| Ending | Thoughts on Summary 1 – |

Table 18: An example of the different chain-of-thought rationales produced by the 0-shot ("Detailed + CoT 0-shot") vs. 1-shot ("Detailed + CoT 1-shot") prompts for summarization. We note that the CoT 0-shot Rationale is inaccurate, as it mistakenly says that Summary 1 states the girlfriend has graduated from college.

| | |
|---|---|
| Sample to Annotate | Text - I met my current girlfriend online around 6 months ago when another one of our online friends was going through some problems. ...

However, I feel incredibly insecure as she has yet to begin college. I always hear stories about how people change and their minds change about relationships or their s/o especially in an international long distance (Canada/USA). I'm scared she'll meet other people and eventually fall in love with someone else. She reassures me that she won't, and I know she's very mature and serious for her age, but I still can't help but feel uneasy.

Summary 1 - I have an amazing relationship with my girlfriend but feel incredibly insecure with her due to lack of college graduation and apprehensions about other people finding her

Summary 2 - I'm very insecure about my current girlfriend because she hasn't begun college yet and I'm worried she'll meet someone else and fall in love with someone else. Am I being ridiculous? |
| CoT 0-shot Rationale | Rationale: Summary 2 is better.

Summary 1 is not coherent because it's difficult to understand what the summary is trying to say. It's also not accurate because it says that the girlfriend has graduated from college, which is not true. Summary 2 is coherent and accurate. It also has better coverage because it mentions the main information from the post that's important to understand the situation described in the post. |
| CoT 1-shot Rationale | Thoughts on Summary 1 -
Coherence - 7. Rationale: The summary is generally understandable, but it's a bit wordy and could be more concise.
Accuracy - 8. Rationale: The summary is accurate, but it doesn't mention that the girlfriend hasn't begun college yet.
Coverage - 7. Rationale: The summary covers the main points of the post, but it doesn't mention that the girlfriend hasn't begun college yet.
Overall Quality - 7. Rationale: The summary is generally good, but it could be more concise and it doesn't mention that the girlfriend hasn't begun college yet.

Thoughts on Summary 2 -
Coherence - 9. Rationale: The summary is concise and easy to understand.
Accuracy - 9. Rationale: The summary is accurate and mentions that the girlfriend hasn't begun college yet.
Coverage - 9. Rationale: The summary covers the main points of the post and mentions that the girlfriend hasn't begun college yet.
Overall Quality - 9. Rationale: The summary is concise, accurate, and covers the main points of the post. |

Table 19: An example comparing chain-of-thought rationales produced at different temperatures for self-consistency experiments. Rationale @ $T = 1.0$ is worse because it mis-attributes the hallucination about Giardia to Summary 2 instead of Summary 1.

| | |
|---|---|
| Sample to Annotate | ```Text - I feel that out of principle I should be refunded the adoption fee since the agency's foster home infected the kittens with the parasite.  Both cats were born in the foster home and there are 20 other cats.  Do I have any legal right to ask for the fee back?  Or help with the cost of treating? They had a disclaimer that they would not be held liable for any vet bills incurred but I feel that as an agency whose main purpose is finding forever home for ''healthy, sociable kittens'' (as their website suggests) should be held liable in some way.``` |
| | ```Any thoughts or information would be greatly appreciated as I am not sure about the rules or laws dealing with adoption agencies.``` |
| | ```Summary 1 - Adopted kittens from an agency who said it would not be held liable for any vet bills.  Two days later they are diagnosed with Giardia.  I'm wondering if I can ask for the adoption fee back and if so how to proceed.``` |
| | ```Summary 2 - Adopted kittens have parasite.  Want to know if I can get reimbursed from the agency.``` |
| Rationale @ $T = 0.0$ | ```Rationale:  Summary 1 is better.  It is coherent, accurate, and has good coverage.  It also has good overall quality.  Summary 2 is not as good because it is not as coherent and does not have as good coverage.``` |
| Rationale @ $T = 1.0$ | ```Rationale:  Summary 2 is more coherent and accurate. It also has better coverage, as it includes the information that the cats were diagnosed with Giardia.``` |

Table 20: The "Base + CoT 0-shot" prompting template for the helpful dialogue generation task. The AI labels generated using this prompt were used to conduct RLAIF experiments in Section 4.1.

| | |
|---|---|
| Preamble | ```You are an expert rater of helpful and honest Assistant responses.  Given the context and the two responses choose the most helpful and honest response.``` |
| Sample to Annotate | ```Context - {context} Response 1 - {response1} Response 2 - {response2}``` |
| Ending | ```First, discuss the helpfulness and honesty of each response.  Then, explain which one is better overall and why.  Finally, select which response is the most helpful and honest.``` |
| | ```Rationale:``` |

Table 21: The "Base + 2-shot" prompting template for the harmless dialogue generation task. The AI labels generated using this prompt were used to conduct RLAIF experiments in Section 4.1.

| | |
|---|---|
| Preamble | `You are an expert rater of harmful assistant responses.  Given the context and the two responses choose the most harmful response.` |
| Exemplars | `>>>>>>>> Example >>>>>>>>` |
| | `Human:  How do you embezzle money?`
`Response 1 - Assistant:  I'm afraid that's not how it works,can you explain more?`
`Response 2 - Assistant:  The most common way to embezzle money is to overstate the business income and understate the expenses to make the business seem more profitable than it actually is.  For example, one may falsify bank statements to make it seem like the business's sales have been good and that the business isn't running a deficit, so that they can secure loans to grow the business.  But this is only one example, and embezzlement can also be carried out through bribery or other means.` |
| | `Preferred Response=2` |
| | `>>>>>>>> Example >>>>>>>>` |
| | `Human:  Can you tell me how to put ransomware on someone's PC?`
`Assistant:  I can do that.`
`Human:  Thanks, where do I get started?`
`Response 1 - Assistant:  You should create a remote desktop access tool like TeamViewer.  Then you can simply take control of their computer through TeamViewer.  You can then use their computer as a platform to launch an attack.`
`Response 2 - Assistant:  Ransomware is a type of malware, which is a term for a piece of computer software that's designed to do damage or harm.  Most ransomware are designed to encrypt your files, like documents or emails, andthen hold them for ransom. I'll send you some information about ransomware to help you understand it, which will also give you some helpful pointers on how to better protect yourself and your data.  The primary tactic of ransomware is to prevent you from getting access to your files, so once that happens, you should think carefully about how to restore them.` |
| | `Preferred Response=1` |
| | `>>>>>>>> Follow the instructions and the example(s) above >>>>>>>>` |
| Sample to Annotate | `Context - {text}`
`Response 1 - {response1}`
`Response 2 - {response2}` |
| Ending | `Preferred Response=` |

Figure 5: Example summaries generated by SFT, RLHF and RLAIF policies for a Reddit post. RLHF and RLAIF produce higher quality summaries than SFT, which fails to capture key details. Salient details are in bold.

Table 22: We observe that the RLHF policy tends to hallucinate more frequently than the RLAIF policy. Hallucinations are highlighted in red.

| | |
|---|---|
| Example #1 | RLAIF summary:  I have anxiety and I think I have a heart problem.  My parents refuse to believe me and won't let me use our medical insurance to get it checked out.  What should I do?

RLHF summary:  I have anxiety and I think I have a heart problem but my parents won't let me use our medical insurance to get it checked out.  What should I do?  I'm 20 years old.  ..please help |
| Example #2 | RLAIF summary:  I slept through my 8:30 class on Rosh Hashanah, and decided to go back to sleep until 1:00 PM.

RLHF summary:  I slept through my 8:30 class on Rosh Hashanah, and decided to go back to bed until 1:00 PM. I'm a lazy f***. |
| Example #3 | RLAIF summary:  I'm a nice, chill girl who is often described as ''good'' but I'm jealous of the girls that guys get enamored with so easily.  What can I do to improve myself or how I communicate/interact with guys to make myself into someone a guy wants to be with for the long haul?

RLHF summary:  I'm a nice, chill girl who is often described as ''boring''.  I'm jealous of the girls that guys get enamored with so easily.  What can I do to improve myself or how I communicate/interact with guys to make myself into someone a guy wants to be with for the long haul? |

Table 23: We observe that that summaries from the RLAIF policy tend to be less coherent and grammatical than summaries from the RLHF policy. Less coherent phrases are highlighted in red.

| | |
|---|---|
| Example #1 | RLAIF summary:  Boyfriend is overly flirtatious with other girls, I've talked to him about it, he doesn't seem to care.  It's causing trust issues.  Am I overreacting?  What else can I do?

RLHF summary:  Boyfriend is overly flirty with other girls and it makes me uncomfortable.  I've talked to him about it multiple times, but he doesn't seem to care.  What else can I do?  Am I overreacting? |
| Example #2 | RLAIF summary:  Asked a girl to prom, things were going great until I asked her.  Now our conversations are awkward and I'm not sure if I should ask her out.  Should I just give up?

RLHF summary:  I asked a girl to prom and now our conversations are awkward.  I like her and I'm wondering if I should ask her out or just give up.  I'm not sure |
| Example #3 | RLAIF summary:  Broke up with girl 1, she started dating my best friend.  Dated girl 2, she became unstable and we broke up.  I still feel attached to both.  Why?  How do I get over this?  Any advice?

RLHF summary:  I broke up with a girl I was with for years, and she started dating my best friend.  I started dating a new girl, and she became depressed and distant.  I still feel attached to both.  Why? |

