# OpenReview forum: "RLAIF: Scaling Reinforcement Learning from Human Feedback with AI Feedback"
_ICLR.cc/2024/Conference — Submitted to ICLR 2024_

### Official Review · Reviewer_LzbJ · 2023-10-31

**Soundness:** 4 excellent
**Presentation:** 4 excellent
**Contribution:** 4 excellent
**Rating:** 8
**Confidence:** 4

**Summary:**

This paper compares Reinforcement Learning from AI-generated intermediate Feedback (RLAIF) with RLHF in summarization and dialog generation tasks. It also investigates techniques to improve AI-generated preference alignment.

**Strengths:**

1. The paper is well-organized, making it easy to follow.
2. It demonstrates RLAIF’s comparability to RLHF in specific tasks and provides optimal settings, offering a more cost-effective solution for AI alignment—a significant contribution given the experiment’s high cost and urgency.

**Weaknesses:**

1. The study only uses non-public "palm2" models, reducing its credibility. Including open-source models could strengthen its validity.
2. The tasks are confined to summarization and dialog generation. Exploring additional areas like QA, code generation, or translation could provide a more comprehensive understanding of AI and human feedback interactions.

**Questions:**

Incorporating an exploration of widely used algorithms like Proximal Policy Optimization (PPO) could enrich the study’s findings.

---

> ### Author Response · Authors · 2023-11-15
> **Reply to Reviewer LzbJ**
>
> Thank you for your thoughtful comments and time spent reviewing our paper. We address each point below:
>
> __Public Availability of PaLM 2 Models__
>
> Thank you for raising the concern regarding the availability of PaLM 2 models. PaLM 2 models are publicly available through Google’s PaLM API (https://developers.generativeai.google/guide/palm_api_overview) and Vertex AI (https://cloud.google.com/vertex-ai/docs/generative-ai/learn/models), which enhances the reproducibility of our results. We will explicitly mention this in the revised paper to ensure clarity for readers.
>
> __Experimenting on More Tasks__
>
> We conduct experiments on two more tasks to further strengthen our findings - specifically on question/instruction answering and harmless dialogue generation. For question/instruction answering, we find that RLAIF and RLHF both outperform the SFT model with 54% win rates (statistically significantly different from 50%). For harmless dialogue generation, we find that the RLAIF and RLHF outperform the supervised baseline, with SFT, RLHF, and RLAIF responding with harmless replies 64%, 76%, and 88% of the time, respectively. Across all 4 tasks, we find that RLAIF achieves comparable results to RLHF.
>
> We acknowledge that there are many more tasks that could be investigated, and we believe that the positive results on these four tasks provide a promising indication that RLAIF can achieve performance comparable to RLHF.
>
> __Additional contributions to the paper__
>
> To enhance the contribution of this work to the research community, we have conducted several additional experiments that expand the scope of our investigation. Specifically:
> * We have compared RLHF and RLAIF on a new task: harmless dialogue generation using Anthropic's Helpful/Harmless preference dataset
> * We have also investigated the performance of RLAIF with an AI labeler of the same size as the policy
> * We have experimented with bypassing the reward model by directly using feedback from a LLM
> * We have conducted experiments combining RLHF and RLAIF
>
> These additional experiments will be incorporated into the revised paper, providing a more comprehensive and impactful contribution to the field.

---

> > ### Author Response · Authors · 2023-11-23
> > **Follow-up: update to paper**
> >
> > We have updated our paper and would like to point out the changes most relevant to your feedback:
> >
> > 1) Appendix D "LLM Labeling Details" - We add a reference to the public versions of PaLM 2
> > 2) Section 4.1 "RLAIF vs RLHF" - we show that RLAIF outperforms RLHF on the harmless dialogue generation task. We also report results on combining RLAIF + RLHF
> >
> > Additionally, we highly encourage you to review the updated version of our paper. As mentioned in our previous comment, we have added significant new contributions that we believe are very valuable to the shared knowledge of the research community. The biggest highlights:
> >
> > * Section 4.2 "Towards Self-improvement" - we show that RLAIF can improve upon the SFT baseline even when the AI labeler is the same size as the policy
> > * Section 4.3 "Direct RLAIF" - we show that directly prompting the LLM for reward scores during RL outperforms the canonical RLAIF setup where AI preferences are distilled into a RM
> >
> > If you have additional concerns, we are eager to hear them.

---

### Official Review · Reviewer_RhkW · 2023-11-01

**Soundness:** 2 fair
**Presentation:** 3 good
**Contribution:** 2 fair
**Rating:** 3
**Confidence:** 4

**Summary:**

This paper compares the efficacy of reinforcement from human feedback (RLHF) and reinforcement from AI feedback (RLAIF) on the PaLM 2 XS model, for the tasks of summarization and "helpful dialogue generation" (Bai, et al. 2022). The paper found that when using a PaLM 2 Large model as the AI labeler, the performance of RLHF and RLAIF were similar. The paper also includes a number of experiments on the use of chain-of-thought in the AI labeler, the size of the AI labeler, and the number of feedback examples in RLHF/RLAIF.

**Strengths:**

The paper is generally clear and well-written, although see below for some framing suggestions. The analysis of the amount of human/AI feedback needed to reach maximum performance is interesting, as is the effect of chain-of-thought and self-consistency on the alignment between AI and human annotations. The qualitative analysis also hints at an interesting topic, namely that RLHF and RLAIF-trained models may be optimizing slightly different objectives; however, it does not give a full treatment to this topic (again, see below for more comments).

**Weaknesses:**

It should be unsurprising that a weaker model (PaLM 2 XS) can be improved based on feedback from a larger model (PaLM 2 L). As noted by the authors in Section 2.2, this can be viewed as a distillation result. While I believe the paper still has some valuable insights, I wish it would be more clear upfront (e.g., in the abstract) that the RLAIF setting described within is one where the target model and the labeling model differ in size.

The qualitative analysis in Section 5 is relatively shallow and would benefit from some additional justification: (1) can you provide more conclusive evidence that these trends exist, e.g,. with human labelers?; and (2) can you provide a hypotheses as to why these trends occur? For example, given that the paper notes only 78% agreement between human and AI labelers, further effort could be put into distinguishing between the labels used to train RLHF and RLAIF, which could elucidate downstream differences in trained model behavior.

The paper also runs a number of experiments to determine whether chain-of-thought reasoning and self-consistency improve alignment between human and AI labelers, finding that self-consistency does not lead to improvements. However, given that human ratings are (1) subjective and (2) subject to noise, the paper should more seriously consider the possibility that lower "AI Labeler Alignment" may not necessarily lead to worse downstream performance. This is partially discussed in Section 4.6, but the authors consider only a single comparison and do not directly compare the two model outputs, instead comparing both of them individually to a supervised fine-tuned baseline.

A minor note: I find it strange that this paper is titled "RLAIF" when it is not the first work to use or introduce the term. See for example Bai, et al. 2022 ("Constitutional AI") for earlier usage of this term. I would recommend the authors remove the title and simply use the subtitle

**Questions:**

N/A

---

> ### Author Response · Authors · 2023-11-16
> **Reply to Reviewer RhkW**
>
> Thank you for your time and for reviewing our paper carefully. We greatly appreciate your suggestions and address them to the best of our ability below:
>
> __Contribution Value of this Work__
>
> We have conducted an additional experiment in which the AI labeler matches the size of the policy and RM. Here, RLAIF surpasses the initial SFT policy, achieving a win rate of 68%. We hope this addresses your initial concern, and we have incorporated this into Sections 3.2 and 4 of the revised paper.
>
> We also believe that our experiments with the larger AI labeler remain valuable to the research community. We agree that it is not entirely unexpected that a weaker model improves based on feedback from a larger model, given that RLHF improves a model with feedback from humans, which can be viewed as a super-capable model. However, the significance of our work lies in quantifying the performance gap between AI and human feedback. Our findings, which indicate that directly replacing human feedback with LLM feedback can achieve comparable results to RLHF, are novel and have not been explored in previous work. We believe this is a valuable contribution to the research community.
>
> As you have suggested, we will also make it clearer that we have two scenarios - one where the AI labeler is larger than the policy, and another where it is the same size.
>
> __Qualitative Analysis__
>
> In response to your concerns regarding the qualitative analysis, we have implemented three changes.
>
> First, we have initiated human evaluation to assess whether the observed trends exist at scale. If we find evidence of the trends, we will provide hypotheses as to why these trends occur. The results of this evaluation will be incorporated into the revised Section 5.
>
> Second, to reflect the explorative nature of this analysis, we have renamed Section 5 from "Qualitative Analysis" to "Qualitative Observations." This emphasizes that the findings represent observations rather than definitive patterns.
>
> __Impact of AI Labeler Alignment on Downstream RL Policy__
>
> To address your concern regarding the lack of direct comparison between the two model outputs in Section 4.6 “End-to-End Sensitivity to AI Labeler Alignment”, we have added a head-to-head evaluation of the win rates of the two RLAIF policies. We find that the policy trained with higher preference label alignment (78% alignment) achieves a win rate of 59% over the policy trained with lower alignment (76% alignment). We will incorporate these results into the revised paper.
>
> We also acknowledge the inherent subjectivity and noise associated with human preference ratings. To mitigate this issue, we employed three raters to evaluate each rating instance. Additionally, the prompt provided to the AI preference labeler closely mirrors the instructions given to human evaluators, leading us to expect a strong correlation between AI Labeler Alignment and the final policy's performance.
>
> Despite these measures, we recognize the possibility that higher AI Labeler Alignment may not necessarily translate to improved downstream performance. Due to the resource-intensive nature of conducting these experiments and human evaluation, we do not intend to conduct extensive studies to fully explore this relationship, which is deserving of an entire study on its own. However, we will clearly acknowledge that we cannot definitively conclude that higher AI Labeler Alignment consistently leads to better downstream performance.
>
> __Framing of Paper__
>
> To make it explicit that we did not introduce RLAIF, we have amended the Abstract to directly credit Bai et al.. In the second sentence of the abstract, we write, “RL from AI Feedback (RLAIF) is an alternative solution introduced by Bai et al. that generates preferences using an off-the-shelf LLM in lieu of human annotators.”
>
> Given that our work is the first to study RLAIF in such depth, we believe that a fair compromise is to leave “RLAIF” in the title while updating the abstract to make it abundantly clear that RLAIF was first introduced elsewhere.
>
>
> __Additional Contributions to the Paper__
>
> To enhance the contribution of this work to the research community, we have conducted several additional experiments that expand the scope of our investigation. Specifically:
>
> * We have compared RLHF and RLAIF on a new task: harmless dialogue generation using Anthropic's Helpful/Harmless preference dataset
> * We have also investigated the performance of RLAIF with an AI labeler of the same size as the policy
> * We have experimented with bypassing the reward model by directly using feedback from a LLM
> * We have conducted experiments combining RLHF and RLAIF
>
> These additional experiments will be incorporated into the revised paper, providing a more comprehensive and impactful contribution to the field.
>
>
> __Final Word__
>
> If we have adequately addressed your concerns, we would greatly appreciate a re-evaluation of your initial rating. Otherwise, please let us know your additional concerns.

---

> > ### Author Response · Authors · 2023-11-23
> > **Follow-up: update to paper**
> >
> > We have updated our paper and would like to point out the changes most relevant to your feedback:
> >
> > 1) Section 4.2 "Towards Self-improvement" - we show that RLAIF can improve upon the SFT baseline even when the AI labeler is the same size as the policy
> > 2) Section 5 "Qualitative Observations" - we updated this section to avoid making any claims beyond what we immediately observe. We defer making conclusive statements about RLAIF and RLHF until we finish our human evaluation to identify these trends, which is still in progress
> > 3) Section 4.4 "Prompting Techniques" - we explicitly acknowledge the possibility that improving AI Labeler Alignment may not lead to downstream gains. Furthermore, we caveat that our "end-to-end sensitivity to AI labeler alignment" experiment suggests the two are correlated, but is by no means conclusive
> > 4) Abstract - we update the abstract to explicitly credit Bai et al (Constitutional AI) for first introducing RLAIF
> >
> > Additionally, we highly encourage you to review the updated version of our paper. As mentioned in our previous comment, we have added significant new contributions that we believe are very valuable to the shared knowledge of the research community. The biggest highlights:
> >
> > * Section 4.1 "RLAIF vs RLHF" - we show that RLAIF outperforms RLHF on the harmless dialogue generation task. We also report results on combining RLAIF + RLHF
> > * Section 4.3 "Direct RLAIF" - we show that directly prompting the LLM for reward scores during RL outperforms the canonical RLAIF setup where AI preferences are distilled into a RM
> >
> > If you have additional concerns, we are eager to hear them. If we have adequately addressed your concerns, we would appreciate if consider re-evaluating your initial rating.

---

### Official Review · Reviewer_BDUS · 2023-11-01

**Soundness:** 3 good
**Presentation:** 3 good
**Contribution:** 4 excellent
**Rating:** 6
**Confidence:** 4

**Summary:**

This paper presents and studies RLAIF, an alternate to RLHF wherein the preference data is synthetically produced by an LLM.

The preference labeling is done by prompting a Palm2-L model, with a prompt that consists of (i) a base/detailed preamble, (ii) optional exemplars, (iii) sample (context + 2 responses) and (iv) ending string. The label is obtained by considering the logprobs for 1 and 2 (after "Preferred Response="). When generating preference labels, the paper also considers (i) CoT reasoning, (ii) self consistency, (iii) mitigating position bias by doing two inference passes.

An RM is trained on the generated preference data, and used to train an LM via RL. There are 3 evaluations: (1) AI labeler alignment which measures the accuracy of the synthetic AI-generated preference data against human preferences, (2) pairwise accuracy of the RM and (3) the human winrate of the RLAIF-trained LMs.

Experiments are conducted on two domains: summarization (tldr) and helpful dialog (anthropic hh-rlhf). There are several takeaways, listed below:

(1) For preference labelling: CoT helps across both domains, inconsistent results between detailed and base preamble, size of the AI labeler helps. Self-consistency (higher temperature) and few-shot exemplars hurt performance.

(2) The RM converges faster (relative to human preferences) with AI generated feedback

(3) RLAIF and RLHF both outperform an SFT model with a winrate of 70% (summarization) and 60% (helpful dialog). RLAIF vs RLHF has a winrate of 50%, suggestion equal performance.

**Strengths:**

This paper presents a comprehensive study of the role of LLM-generated feedback in RLHF. By performing sound experiments at each stage of the RLHF process, this paper shows RLAIF to be a reliable alternative to human preferences: (1) the agreement between the human preference data vs AI preference data (with multiple approaches), (2) the performance of the RM trained on different data and (2) the human evaluation performance of the LLMs trained with RLHF vs RLAIF. Though the methodology may not be novel, the comprehensive experiments are insightful and valuable to the community.

This paper presents several valuable and insightful results (1) impact of CoT/self consistency during preference labeling, (2) impact of the AI labeler size on accuracy, (3) RM performance as a function of amount of preference data, (4) studying position bias.

**Weaknesses:**

Since the comprehensive experimentation is a strength of this paper, is it imperative that the experiments and analysis is sound. The following points would benefit from additional experimentation or discussion:

(1) In Figure4, it's not clear to me why adding exemplars hurts performance. There is a one sentence justification for this on page6, but I think it's insufficient, since exemplars hurt performance even without CoT. Is it the case that your exemplars are low quality? Could this be mitigated by using the 0-shot generations as exemplars? It would also be valuable to understand the role of exemplars at different labeler sizes (e.g. P2-S using exemplars produced by P2-L).

(2) Again, regarding Figure4: Since human annotators have a 60-75% agreement rate on these datasets, I wonder if small differences in accuracy in Table4 are meaningful. Is it possible that the AI labels are more correct than the human preferences? Analyzing the disagreements may shed light on this. If so, what does it imply about the results/takeaway in Figure4.

(3: suggestion) This is not a weakness, but more of a suggestion. It would be interesting to see the relationship between Figure4 and Figure5b. Does a higher quality AI-preference dataset necessarily lead to a more accurate RM?

(4) The human evaluators used to assess the final RLHF/RLAIF/SFT-trained LLMs are distinct from the preference data/RM. How do we know if these annotators are modeling the same preference policy expressed by the datasets/RMs, and not (for example) just picking the longest response? To this end, it would be good to either show RM scores for the final LLMs or to measure the agreement of the human evaluators against the original preference data or the RMs.

**Questions:**

Questions in the weakness section:

1. Why do exemplars hurt performance of the preference labeler?
2. Is it possible that the AI labels are more correct than the human preferences? If so, what does it imply about the results/takeaway in Figure4?
3. Does a higher quality AI-preference dataset necessarily lead to a more accurate RM?
4. Does a higher quality/more accurate RM necessarily lead to a better/more preferred LLM?
5. How do we know if the human annotators are modeling the same preference policy expressed by the datasets/RMs, and not (for example) just picking the longest response?

---

> ### Author Response · Authors · 2023-11-16
> **Reply to Reviewer BDUS**
>
> We sincerely appreciate your time and insightful feedback on our paper. We value your thoughtful suggestions and address each point in detail below:
>
> __The Effect of In-Context Learning on AI Labeler Alignment__
>
> We have also conducted experiments on harmless dialogue generation (see last part of reply), where we found that adding exemplars increases AI Labeler Alignment. Our observations suggest that the optimal prompting techniques are task-specific.
>
> We do not believe that low-quality exemplars are to blame for ICL not working. For all tasks, we carefully handpicked high quality exemplars representative of the preference task. We also measured alignment for "Base 1-shot" on summarization using 10 random exemplars, yielding a maximum/minimum alignment of 76.1%/74.4%, none surpassing the 76.1% alignment of "Base 0-shot". We will add these details to the paper.
>
> Interestingly, existing literature suggests that the role of exemplars and chain-of-thought prompting is still not fully understood (e.g., [1] and [2] show that corrupting exemplars and chain-of-thought surprisingly yield performance gains). We agree that deeper understanding of exemplars is an important research area.
>
> [1] Rethinking the Role of Demonstrations: What Makes In-Context Learning Work? - https://aclanthology.org/2022.emnlp-main.759.pdf
>
> [2] Towards Understanding Chain-of-Thought Prompting: An Empirical Study of What Matters - https://arxiv.org/pdf/2212.10001.pdf
>
> __Whether Differences in AI Labeler Alignment are Meaningful__
>
> We studied the sensitivity of RL-trained policies to the AI labeler accuracy, revealing that a small disparity can lead to a significant difference in win rates. Specifically, a policy trained with preference labels achieving 78% AI Labeler Alignment outperforms one trained with preference labels achieving 76% alignment, securing a win rate of 59% (statistically significantly above 50%). We will report these results in Section 4.6 "End-to-End Sensitivity to AI Labeler Alignment”.
>
> While we cannot definitively conclude that higher AI Labeler Alignment consistently translates to superior RL policies from this one experiment, this result suggests that such a correlation at least exists in certain cases. Due to the expensive nature of conducting these experiments and human evaluation, we agree that further investigations are necessary and defer more comprehensive study to future work.
>
> __AI Labeler Alignment vs. RM Accuracy__
>
> We acknowledge the importance of thoroughly understanding the relationship between AI Labeler Alignment and RM accuracy. To address this concern, in the revised paper, we will include the RM accuracies for the two RMs trained on different AI labels in Section 4.6 "End-to-End Sensitivity to AI Labeler Alignment.”
>
> We also would like to note that training and evaluation for all RMs are done in the exact same way.
>
> __Is Human Eval Aligned with Preference Datasets?__
>
> We acknowledge the potential for discrepancies between the human raters who curated the preference datasets and the individuals we engaged for evaluation. Additionally, differences in training or instruction may introduce further variations. To minimize the impact of such mismatches, we meticulously replicated the settings provided in the original papers. Also, as you pointed out, it is possible that the AI labeler may provide more accurate assessments than the humans who initially rated the dataset in some cases.
>
> Incorporating an agreement analysis between our human evaluators and the preferences in the preference datasets is an excellent suggestion. If we can afford additional human evaluation, we will integrate this analysis into the camera-ready version to further substantiate the value of our human annotators' feedback.
>
> Regarding the possibility that annotators pick the longest response, our post-hoc analysis suggests that human evaluators continue to favor RLAIF and RLHF model outputs even after controlling for response length. This post-hoc analysis is similar to the approach employed by Stiennon et al. in their work "Learning to Summarize from Human Feedback" (https://arxiv.org/pdf/2009.01325.pdf).
>
> __Additional contributions to the paper__
>
> To enhance the contribution of this work to the research community, we have conducted several additional experiments that expand the scope of our investigation. Specifically:
> * We have compared RLHF and RLAIF on a new task: harmless dialogue generation using Anthropic's Helpful/Harmless preference dataset
> * We have also investigated the performance of RLAIF with an AI labeler of the same size as the policy
> * We have experimented with bypassing the reward model by directly using feedback from a LLM
> * We have conducted experiments combining RLHF and RLAIF
>
> These additional experiments will be incorporated into the revised paper, providing a more comprehensive and impactful contribution to the field.

---

> > ### Author Response · Authors · 2023-11-23
> > **Follow-up: update to paper**
> >
> > We have updated our paper and would like to point out the changes most relevant to your feedback:
> >
> > 1) Section 4.4 "Prompting Techniques" - we share evidence for why we believe low-quality exemplars are not to blame for ICL not improving results
> > 2) Appendix N "End-to-end Sensitivity to AI Labeler Alignment" - we report end-to-end results on how differences in AI labeler alignment translate to downstream changes in performance
> > 3) Appendix H "Reward Model Accuracy" - we report the RM accuracies of all RMs used in RLAIF and RLHF across the paper. This includes two RMs trained from labels with different degrees of AI labeler alignment. We do not find a strong relationship between AI labeler alignment and RM accuracy
> > 4) Appendix J "Controlling for Response Length" - we report results after controlling for response length and find that evaluators are not simply picking the longest responses
> >
> > Additionally, we highly encourage you to review the updated version of our paper. As mentioned in our previous comment, we have added significant new contributions that we believe are very valuable to the shared knowledge of the research community. The biggest highlights:
> >
> > * Section 4.1 "RLAIF vs RLHF" - we show that RLAIF outperforms RLHF on the harmless dialogue generation task. We also report results on combining RLAIF + RLHF
> > * Section 4.2 "Towards Self-improvement" - we show that RLAIF can improve upon the SFT baseline even when the AI labeler is the same size as the policy
> > * Section 4.3 "Direct RLAIF" - we show that directly prompting the LLM for reward scores during RL outperforms the canonical RLAIF setup where AI preferences are distilled into a RM
> >
> > If you have additional concerns, we are eager to hear them. If we have adequately addressed your concerns, we would appreciate if consider re-evaluating your initial rating.

---

### Official Review · Reviewer_4rk2 · 2023-11-01

**Soundness:** 2 fair
**Presentation:** 3 good
**Contribution:** 3 good
**Rating:** 6
**Confidence:** 4

**Summary:**

This paper aims to analyze the performance of reinforcement learning with AI feedback (RLAIF). RLAIF is similar to RLHF but instead of collecting expensive human annotations, the preferences are generated by another LLM. Under the setup of this paper, the RLAIF achieves a similar win rate as RLHF in human evaluation, showing that RLAIF can potentially mitigate the scalability issue of RLHF.

**Strengths:**

* Investigating RLAIF’s performance, especially comparing it to RLHF, is important and timely.
* The achievement of RLAIF in this paper’s setup is interesting. It can achieve the same level of performance as RLHF.
* The writing is really clear.

**Weaknesses:**

* If I understand correctly, both RLAIF and RLHF are based on the SFT baseline (fine-tuned PaLM2 XS) and REINFORCE. In this case, the key difference among RLAIF and RLHF in the experiments is the used reward models and the training data for the reward models. Therefore questions raise:
  * What is the accuracy of the finally trained Human Feedback RM? I only see in Appendix E that the RM “is trained until the training loss and accuracy curves plateau”, but in Figure 5(b) it is still not plateau. Having this number can help readers understand (1) if the on-par performance of RLAIF and RLHF is due to using RMs with similar accuracy.
  * To further dig into the above question, analysis of how RM accuracy affects the RLHF results can also be helpful.
  * The AI feedback quality is the base and will first affect the trained AI feedback RM and then the RM will affect the result. However, the analysis in Section 4.6 entangles the two steps. It causes confusion if the performance difference comes from the trained RM or the AI feedback with different AI Labeler Alignment?
  * Moreover, only the AI feedback with 76.1% and 78% AI Labeler Alignments are compared, if having a wider range analysis, it will be easier to understand the impact of the AI Labeler Alignments to the trained RM.
* About human evaluation. Since the reported number is only the total human rating, I’m curious about other statistics. How many input-outputs examples are used? How many evaluators rate the same input-outputs example? What’s the inter-annotator agreement of the results?
* The experimental results after controlling the length in Appendix F only shows the comparison of RLAIF vs SFT and RLHF vs SFT. Is there also a comparison between RLAIF vs RLHF?

**Questions:**

* About the experimental setup,
  * Are the SFT baselines using only the preferred responses in the datasets? I have checked section 3.3, appendix A.1 and E, but haven’t seen the answer to this question.
  * Is there a reason why the reward models are also initialized from the SFT models (described in Section 3.3)? Their output spaces are different. Is it a random try or based on some statistics?
  * What is the baseline used for REINFORCE (mentioned as “we use REINFORCE with a baseline” in Section 3.3)? Is it the output of the value model in the authors’ setup?
* Presentation suggestion:
  * The paper describes in-context learning with exemplars (section 2.1) and self-consistency (section 2.1.3) as they are a part of the used methodology. However, in experiments (Figure4), they turn out useless and not applied in the end. In this case, I would suggest not to put much emphasis on them but only mention them and say the authors also study their effects.
  * Add that Figure 5 (a)(b) are results on summarization task in the caption.
* In section 4.2, “we observe that the optimal configuration employs chain-of-thought reasoning and no in-context learning (“Detailed + CoT 0-shot”)” Should here be “Detailed / Base + CoT 0-shot”, since for summarization the best is detailed and for helpfulness the best is base?

---

> ### Author Response · Authors · 2023-11-16
> **Reply to Reviewer 4rk2**
>
> Thank you for your time and for reviewing our paper carefully. We greatly appreciate your thoughtful and detailed suggestions. We address your comments below:
>
> __Relationship Between AI Labeler Alignment, RM Accuracy, and Downstream RL Policy__
>
> We acknowledge the importance of thoroughly understanding the relationship between RM accuracy and downstream RL results. To address this concern, in the revised paper, we will include the RM accuracies for both human feedback RMs and AI feedback RMs in Section 4.1. Additionally, we will report the RM accuracies for the two RMs trained on different AI labels in Section 4.6, titled "End-to-End Sensitivity to AI Labeler Alignment.”
>
> While we recognize the importance of understanding the relationship between RM accuracy and downstream RL results, we believe this is a broad research question deserving a comprehensive study of its own. Additionally, the nature of RL fine-tuning and human evaluation demands substantial resources, making an in-depth exploration of this relationship outside the scope of our work. Nevertheless, we recognize the significance and potential of this research direction.
>
> We would also like to emphasize that all RMs are trained following a standardized procedure. This consistency in training procedure of the RM does not favor any specific approach, allowing for a fair evaluation of the different methods employed.
>
>
> __Human Evaluation Statistics__
>
> To answer your questions about human evaluation, we constructed a set of 2k rating instances across all of our RL experiments, which includes evaluations for additional experiments that we are adding to the revised paper (see last section of this reply). Each instance comprised a single context and three distinct model responses (e.g., RLAIF, RLHF, SFT), resulting in a total of 6k unique input-output pairs subjected to human evaluation. Additionally, each instance was assessed by three independent raters, which we will use to calculate inter-annotator agreement. These details will be incorporated into Section 3.4 "Human Evaluation" of the revised paper to enhance the transparency of our evaluation methodology.
>
> __Length-Controlled Results__
>
> You raise an excellent point about calculating the length-controlled results for RLHF vs. RLAIF directly. We will calculate this quantity and incorporate the result in Appendix F.
>
>
> __Other Questions__
>
> > Are the SFT baselines using only the preferred responses in the datasets?
>
> We do not conduct SFT on the preferred responses. For summarization, we solely train on the supervised data from the TL;DR dataset (note that this is distinct from the TL;DR *preference* dataset). For other tasks, we do not conduct SFT at all. We will make this clear in the revised paper.
>
> > Is there a reason why the reward models are also initialized from the SFT models (described in Section 3.3)? Their output spaces are different. Is it a random try or based on some statistics?
>
> We initialize the RM from the SFT model because we have found that initializing the RM from the SFT model improves RM performance. The hypothesis behind this is that the RM must be able to adapt to the scoring task, and the SFT model adds more domain-specific training. This is also an observation shared by other researchers - see https://arxiv.org/pdf/2009.01325.pdf, https://notesonai.com/RLHF+-+Reinforcement+Learning+with+Human+Feedback, and https://huyenchip.com/2023/05/02/rlhf.html.
>
> > What is the baseline used for REINFORCE (mentioned as “we use REINFORCE with a baseline” in Section 3.3)? Is it the output of the value model in the authors’ setup?
>
> As you mentioned, we use the value function as the baseline in REINFORCE. See Appendix C for details.
>
> Lastly, we will implement your recommendations on “Presentation suggestion” and fix the point regarding section 4.2. Thank you for pointing these out.
>
>
> __Additional contributions to the paper__
>
> To enhance the contribution of this work to the research community, we have conducted several additional experiments that expand the scope of our investigation. Specifically:
> * We have compared RLHF and RLAIF on a new task: harmless dialogue generation using Anthropic's Helpful/Harmless preference dataset.
> * We have also investigated the performance of RLAIF with an AI labeler of the same size as the policy
> * We have experimented with bypassing the reward model by directly using feedback from a LLM
> * We have conducted experiments combining RLHF and RLAIF
>
> These additional experiments will be incorporated into the revised paper, providing a more comprehensive and impactful contribution to the field.

---

> > ### Author Response · Authors · 2023-11-23
> > **Follow-up: update to paper**
> >
> > We have updated our paper and would like to point out the changes most relevant to your feedback:
> >
> > 1) Appendix H "Reward Model Accuracy" - We report the accuracy for all RMs and add our thoughts on the relationship between RM accuracy and RL results
> > 2) Appendix I "Human Evaluation Details" - We add in statistics about human evaluation, answering your questions of "How many input-outputs examples are used? How many evaluators rate the same input-outputs example? What’s the inter-annotator agreement of the results?"
> > 3) Appendix J "Controlling for Response Length" - We add in the length-controlled results of comparing RLAIF vs RLHF
> > 4) Appendix G "Model Training Details" - We make it explicit what datasets the SFT baselines are trained on. We justify our choice of initializing the RM from the SFT model vs. directly from PaLM 2 XS
> > 5) Appendix F "REINFORCE for Language Models" - This was already in the previous version, but here you can find the formal description of the value function baseline we use for REINFORCE
> > 6) Appendix M "Self-consistency" - Following your suggestion, we move self-consistency results to the Appendix. We also updated Section 4.4 with more details on in-context learning
> >
> > Additionally, we highly encourage you to review the updated version of our paper. As mentioned in our previous comment, we have added significant new contributions that we believe are very valuable to the shared knowledge of the research community. The biggest highlights:
> >
> > * Section 4.1 "RLAIF vs RLHF" - we show that RLAIF outperforms RLHF on the harmless dialogue generation task. We also report results on combining RLAIF + RLHF
> > * Section 4.2 "Towards Self-improvement" - we show that RLAIF can improve upon the SFT baseline even when the AI labeler is the same size as the policy
> > * Section 4.3 "Direct RLAIF" - we show that directly prompting the LLM for reward scores during RL outperforms the canonical RLAIF setup where AI preferences are distilled into a RM
> >
> > If you have additional concerns, we are eager to hear them. If we have adequately addressed your concerns, we would appreciate if consider re-evaluating your initial rating.

---

### Meta-Review · Area_Chair_NRV2 · 2023-12-11

**Metareview:**

From the perspective of AC, this paper lacks sufficient innovation and the experiments are not comprehensive enough. RLAIF is a broad concept, yet the authors' experiments are mainly based on PaLM, without extending to a wider range of open-source models. This raises concerns about the validity of the experiments. Additionally, from the AC's viewpoint, it seems the authors might be over-claiming. As Reviewer RhkW noted
> A minor note: I find it strange that this paper is titled "RLAIF" when it is not the first work to use or introduce the term. See for example Bai, et al. 2022 ("Constitutional AI") for earlier usage of this term. I would recommend the authors remove the title and simply use the subtitle.

I am unclear about the authors' motivation and highly agree with Reviewer RhkW's suggestion, as it shows more respect for the efforts of the authors of 'Constitutional AI.'

During the rebuttal, the authors stated, 'Given that our work is the first to study RLAIF in such depth,' which AC feels is somewhat unfair in claiming to be the 'first.'

Based on the above, I choose to reject this paper as it does not meet the standards required for acceptance at ICLR.

**Justification For Why Not Higher Score:**

1. The experiments, based solely on PaLM, lack convincing power. The reliance on just one model limits the persuasiveness of the results.
2. The authors over-claim significantly. Their work is limited to a single model, with tasks confined to summarization and dialogue generation. The authors' statement, 'Given that our work is the first to study RLAIF in such depth,' is not objective from an AC's perspective.
3. As the reviewers have pointed out, there are numerous questionable aspects in the paper regarding the demonstration of RLAIF's superiority.

**Justification For Why Not Lower Score:**

N/A

---

### Decision · Program_Chairs · 2024-01-16

Reject